# Global spatiotemporal synchronizing structures of spontaneous neural activities in different cell types

Liang Shi [1,2,4], Xiaoxi Fu[1,2,4], Shen Gui[1,2], Tong Wan[3], Junjie Zhuo[3], Jinling Lu [1,2] ✉ & Pengcheng Li [1,2,3] ✉

Increasing evidence has revealed the large-scale nonstationary synchronizations as traveling waves in spontaneous neural activity. However, the interplay of various cell types in fine-tuning these spatiotemporal patters remains unclear. Here, we performed comprehensive exploration of spatiotemporal synchronizing structures across different cell types, states (awake, anesthesia, motion) and developmental axis in male mice. We found traveling waves in glutamatergic neurons exhibited greater variety than those in GABAergic neurons. Moreover, the synchronizing structures of GABAergic neurons converged toward those of glutamatergic neurons during development, but the evolution of waves exhibited varying timelines for different sub-type interneurons. Functional connectivity arises from both standing and traveling waves, and negative connections can be elucidated by the spatial propagation of waves. In addition, some traveling waves were correlated with the spatial distribution of gene expression. Our findings offer further insights into the neural underpinnings of traveling waves, functional connectivity, and resting-state networks, with cell-type specificity and developmental perspectives.

The spontaneous neural activity of the brain demonstrates self-organized intrinsic dynamics[1–7], which are closely associated with stimulation, cognition, and behavior and range from highly synchronized to desynchronized[8]. Functional connectivity (FC) has been commonly employed to depict the spatial organization of brain synchronization and is recognized for its changes during development and across pathological conditions[9–14]. However, mounting evidence gathered in various states, including wakefulness, using various techniques ranging from the cellular to the whole-brain large-scale has revealed the widespread presence of nonstationary synchronization[15–24]. These phenomena may encompass spatiotemporal patterns, standing/traveling waves, and zero-lag/time-lag synchronies and exhibit regional specificity at large scales[15]. Moreover, their regional specificity may not align with brain regions or the structural connectome[25,26]. FC, as a factor of stationary synchronization, is influenced by nonstationary synchronization, suggesting that some phenomena previously interpreted as desynchronization might in fact be nonstationary synchronization.

Although brain synchronization is primarily attributed to glutamatergic excitatory neurons, GABAergic inhibitory interneurons also contribute to this process[27–30]. Several experiments and simulations have shown that one possible mechanism for neural desynchronization is through excitatory and inhibitory neurons sharing the same inputs and outputs[8,31,32]. Fine-tuning large-scale neural activity relies on the intricate interplay between different neuronal types, chiefly glutamatergic and GABAergic neurons, although this interplay remains

[1]Britton Chance Center for Biomedical Photonics and MoE Key Laboratory for Biomedical Photonics, Advanced Biomedical Imaging Facility, Wuhan National Laboratory for Optoelectronics, Huazhong University of Science and Technology, Wuhan, Hubei 430074, China. [2]Research Unit of Multimodal Cross Scale Neural Signal Detection and Imaging, Chinese Academy of Medical Science, HUST-Suzhou Institute for Brainsmatics, JITRI, Suzhou 215100, China. [3]State Key Laboratory of Digital Medical Engineering, School of Biomedical Engineering, Hainan University, Sanya 572025, China. [4]These authors contributed equally: Liang Shi, Xiaoxi Fu. ✉e-mail: lujinling@mail.hust.edu.cn; pengchengli@mail.hust.edu.cn

unclear. During development, the integration of these neuronal populations remains immature, leading to distinct patterns of neural activity at the macroscopic level in young animals. This unique phenomenon offers a prime opportunity for in-depth investigation and analysis of the large-scale synchronization dynamics between these cell types.

In this study, we investigated the spatiotemporal structures and deviations in neural activity within distinct genetically defined cell types during mouse development and across diverse states. We focused on one type of glutamatergic excitatory neuron expressing the second vesicular glutamate transporter (VGLUT2)[30] and three types of GABAergic inhibitory interneurons expressing parvalbumin (PV), somatostatin (SOM or SST), and vasoactive intestinal peptide (VIP)[33,34]. To achieve this goal, we used calcium fluorescence imaging with neuron type-specific Cre driver lines[35] and genetically encoded calcium indicators (GECIs)[36]. We collected calcium fluorescence data from the mouse cortex at three developmental time points (P14, P28, and P56) and three anesthesia/awake states (at rest, during spontaneous movement and under anesthesia). We analyzed "standing" and "traveling" waves identified using complex principal component analysis (CPCA)[15,37] and assessed their association with FC. These standing/traveling waves extracted by CPCA account for much of the global spatial structure found in human resting-state functional MRI as

previously studied[15]. Additionally, we utilized weighted gene coexpression network analysis (WGCNA)[38] to analyze the gene expression data and investigated the relationships between genes and traveling waves. The method we employed, which combined gene analysis with traveling waves, holds great promise for identifying the functional roles of time-lag synchronies. Using fluorescent signals, we also compared the effects of the commonly used but controversial method of global signal regression (GSR) on processing. Our study provides insights into the changes in neural synchronization structures in mice, with significant implications for understanding brain function development and organization, as well as addressing limitations that hinder the interpretation of resting-state functional connectivity data in fMRI studies.

## Results

### Zero-lag and time-lag synchronization structures of spontaneous neural activity in VGLUT2 neurons

We assume that the calcium signal time series obtained from the neural activity can be expressed as the superposition of standing/traveling waves, which are potentially time-delayed in different regions. Therefore, we decomposed the fluorescence signals corresponding to neural activity and extracted a set of standing and traveling waves (Fig. 1) from spontaneous neural activity in VGLUT2 at P56 via CPCA.

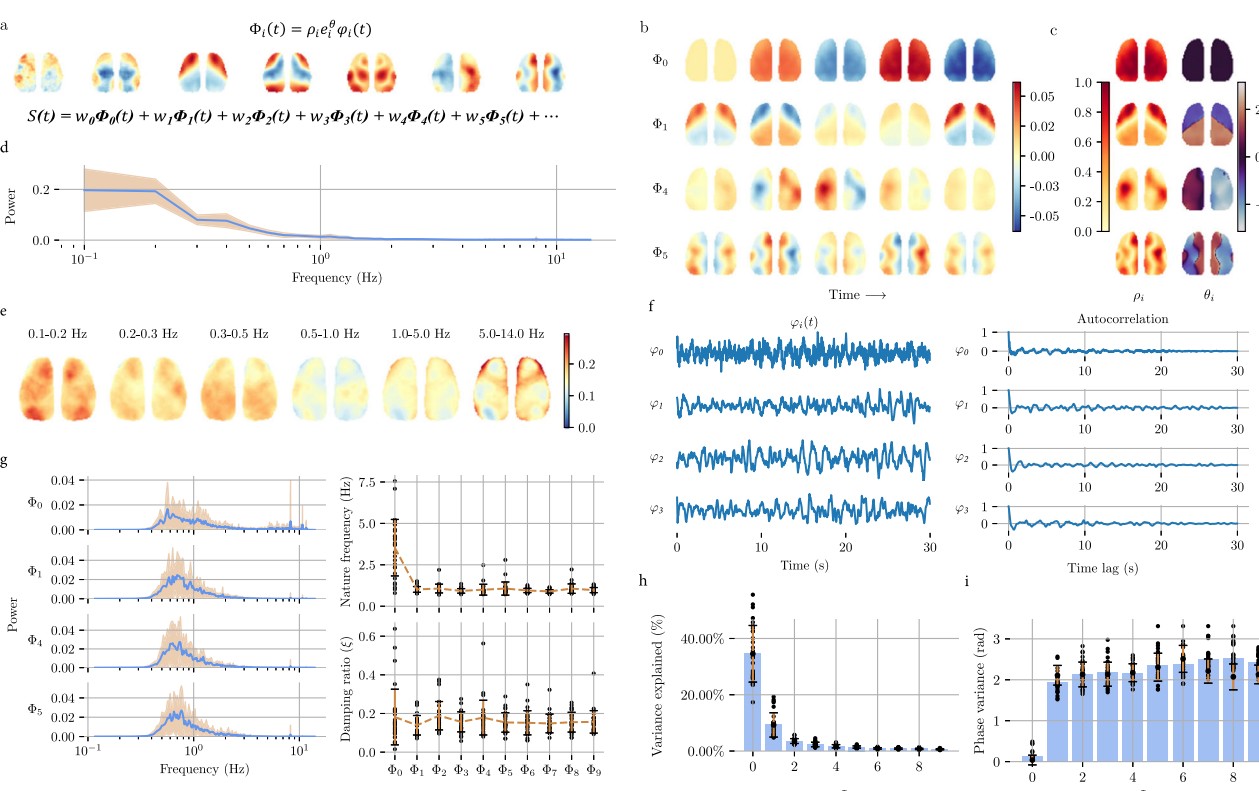

**Fig. 1 | Large-scale neural activity is organized by standing and traveling waves (VGLUT2, P56). a** Fluorescent signals $S(t)$ of neural activity can be decomposed into linear superpositions of standing and traveling waves $\Phi_i(t)$ distributed throughout the cortex; these signals are extracted from spontaneous neural activity in VGLUT2 neurons at P56. The weight $w_i$ is the eigenvalue of $\Phi_i$, $w_i\varphi_i$ gives the principal component "scores", and $(w_i\rho_i e^{\theta_i})/\sqrt{n_{time}-1}$ gives the principal component "loadings". **b** Spatiotemporal patterns (over one cycle) of waves $\Phi_0,\Phi_1,\Phi_4,\Phi_5$, showcasing only significant waves identified in the subsequent text. **c** Spatial distributions of waves $\Phi_0,\Phi_1,\Phi_4,\Phi_5$. $\rho$: Intensity distribution. $\theta$: Phase distribution (time lag) in rad. **d** The power spectral density (PSD) of the fluorescence signal (pixel averaged, normalized to unit energy). **e** Spatial maps of spectral power in 6 nonoverlapping frequency bands, with pixels normalized to unit energy.

**f** Waveforms $\varphi_i(t)$ (first 30 s in 180 s acquisition) and autocorrelation functions of waves $\Phi_0,\Phi_1,\Phi_4,\Phi_5$. **g** Unit-energy PSDs of $\varphi_0,\varphi_1,\varphi_4,\varphi_5$, with interexperimental variability (shaded). Natural frequencies ($f$) and damping ratios ($\xi$) of the 10 strongest waves. The error bars represent the mean ± standard error across the experiments. 23 experiments over $n=10$ male mice. **h** The proportion of the 10 strongest waves, measured by the variance ratio of each wave relative to the original signal. The error bars represent the mean ± standard error across the experiments. 23 experiments over $n=10$ male mice. Source data are provided as a Source Data file. **i** The spatial distribution uniformity of the 10 strongest waves, measured by the circle variance $\theta_i$. The error bars represent the mean ± standard error across the experiments. 23 experiments over $n=10$ male mice. Source data are provided as a Source Data file.

The waves had different proportions in the original signal (Fig. 1a, h) and exhibited varying magnitude and time lags in different brain regions (Fig. 1b, c). The spatial distributions of all the extracted waves (10 waves) are presented in Supplementary Fig. 1. These waves are widely distributed across the cortex and recur in a quasiperiodic manner. This quasiperiodic recurrence is similar to the quasiperiodic patterns (QPPs)[39] or intrinsic oscillatory modes[40] observed in fMRI studies. The wave $\Phi_0$ constituted the largest proportion of the signal, ~35%, with an almost zero time lag across the cortex (Fig. 1c). This suggests a global mode of oscillation with minimal spatial phase variation, which is termed a "standing wave". In contrast, other waves exhibit substantial spatial phase variation, indicating that they are "traveling waves" across the cortex.

Three spatiotemporal patterns of fMRI data were previously observed in resting humans[15]. Standing and traveling waves decomposed from mice seem to share some similar characteristics with those in humans. Specifically, the wave $\Phi_1$ that we identified resembles the first spatiotemporal pattern in humans, which is a switch between negative BOLD amplitudes and positive BOLD amplitudes within the somato-motor-visual (SMLV) complex. Thus, we denoted waves such as $\Phi_1$ as $\Phi_{Sensory-motor}$ or $\Phi_{SM}$. We posit that $\Phi_2$ may also be related to SMLV (Supplementary Fig. 1), so we denoted this parameter as $\Phi_{SM2}$. The $\Phi_3$ and $\Phi_5$ waves of VGLUT2 at P56 (Supplementary Fig. 1) were highly concentrated in the RSP and M2 regions, suggesting that there may be a potential link with the DMN, as proposed in previous studies[41,42]. Thus, we denote waves such as $\Phi_5$ as $\Phi_{DMN-like}$. A comparable phenomenon to the third spatiotemporal pattern in human subjects was also identified. These similarities between spontaneous neural activity in mice and BOLD signal fluctuations in humans suggest the conservation of brain operations across species.

We found that the spatial frequency distribution was uneven, with low frequencies dominating the entire cortex (Fig. 1e). The frequency characteristics of the standing/traveling waves were different from those of the signals, with almost single-peaked distributions. This contrasts with the fluorescent signals, which had a broader frequency distribution. The natural frequency ($f$) reflects the frequency of the energy concentration in the absence of damping. The damping ratio ($\xi$) measures the degree of energy concentration, with smaller values indicating higher concentrations. It is also the reciprocal of the Q-factor ($\xi = 0.5Q^{-1}$). $f$ and $\xi$ were estimated using autocorrelation functions (Fig. 1f, Methods for details). The natural frequencies of each standing/traveling wave in VGLUT2 neurons are similar. The damping ratios $\xi$ of these waves are small (Fig. 1g), indicating a high degree of energy concentration within a narrow band of frequencies. Waves extracted by CPCA were decomposed based on their correlations without considering frequency. However, the results of the decomposition showed frequency specificity, indicating that the organization of long-range correlations between brain regions is frequency dependent, consistent with previous research results[40]. However, $\Phi_0$, the "standing" wave, is unique because it showed a greater degree of inconsistency in frequency characteristics between different experiments and mice.

### Standing/traveling waves explain FC with negative connections due to signal time lags

FC is a crucial and fundamental measure of neural activity synchronization. Due to the properties of CPCA (in which the covariance between different components is 0), the standing and traveling waves obtained by CPCA decomposition are also FC decompositions; that is, $FC = s_0 C_{\Phi_0} + s_1 C_{\Phi_1} + \cdots + s_i C_i + \cdots$, $C_{\Phi_i}$ are correlation matrices calculated from the spatiotemporal patterns of wave $\Phi_i$ (reconstructed FC matrices), as shown in Fig. 1b. As the number of waves increases, the linear superposition of $C_{\Phi_i}$ becomes closer to the original $FC$ and $FC_{GSR}$ (Fig. 2b). Because the waveforms of $\Phi$ are the same at different locations except for the time lag, the differences and even negative

correlations in these reconstructed FC matrices ($C_{\Phi_1}$ ~ $C_{\Phi_4}$ in Fig. 2a) are caused by the time lag. Both FC and standing/traveling waves capture the synchronization features of the brain; therefore, we refer to both as synchronization structures.

Our results demonstrate that the $FC$ matrix and $FC_{GSR}$ matrix are strongly correlated ($r = 0.93 \pm 0.03$); in other words, $FC \approx \alpha FC_{GSR} + \beta$, where $\alpha$ and $\beta$ are constants. The effect of GSR is approximately equivalent to subtracting a constant from FC in a way that essentially does not affect the FC structure, which is more conducive to analysis. The correlation matrices reconstructed from $\Phi_0$ are almost constant matrices ($C_{\Phi_0}$ in Fig. 2a and Supplementary Fig. 3), and the high similarity between the average FC without GSR and the spatial distribution of $\Phi_0$ further indicates that $\Phi_0$ shares characteristics with the "global signal" eliminated by GSR, and removing $\Phi_0$ from the raw signal produces similar outcomes to GSR itself. We suggest that standing waves such as $\Phi_0$ represent zero-lag synchronizations, denoted as $\Phi_{Global}$ or $\Phi_G$. In contrast, traveling waves (for example, $\Phi_1$, $\Phi_4$ and $\Phi_5$) with positive and negative connections in their correlation matrices represent time-lag synchronizations.

The spatial distribution of $\Phi_1$ encompasses two clusters (Fig. 2a, $C_{\Phi_1}$), which are reminiscent of the FC matrix (Fig. 3a) and the spatial distribution of the average $FC_{GSR}$ (Fig. 3c). Wave $\Phi_i$ contributes the most to the overall trend of $FC$ and $FC_{GSR}$ (Fig. 2b). Notably, the phase difference in $\pi$ between these two regions indicated a peak-to-valley pattern (Fig. 1b, c), which suggested a negative correlation (Fig. 2a). The distribution pattern of $FC_{GSR}$ on the cortex can be divided into two internally correlated modules (FC post-GSR in Fig. 2a, $D_{FC_{GSR}}$ in Fig. 2c) and aligns with the spatial distribution of $\Phi_1$ (Fig. 2d). The distribution of the phase of $\Phi_1$ almost completely explains the sign of $FC_{GSR}$ (whether a connection is positive or negative).

Moreover, the seed connections of FC within the ipsilateral cortex are typically stronger than the connections with the contralateral cortex. This difference in functional connectivity across hemispheres varies spatially, as demonstrated in Fig. 2c. We found that this asymmetry phenomenon is akin to $\Phi_4$ (Fig. 2d). Thus, we denote waves such as $\Phi_4$ as $\Phi_{Transhemispheric}$ or $\Phi_T$. All the reconstructed FC maps from waves $\Phi_G$, $\Phi_{SM}$, $\Phi_T$ and $\Phi_{DMN-like}$ are presented in Supplementary Fig. 3.

### Similarity of synchronization structures in four types of neurons at P56

Except for a few regions, the three types of GABAergic neurons developed $FC$ and $FC_{GSR}$ patterns similar to those of VGLUT2 at P56 (Fig. 3a, c). However, the connection strength of VGLUT2 was significantly greater than that of GABAergic neurons (Fig. 3b). The degree of FC similarity post-GSR between GABAergic neurons and VGLUT2 at P56 varies spatially across the cortex, as shown in Fig. 3g, and changes over time, increasing with mouse age (Supplementary Fig. 2a, b).

Traveling waves revealed marked differences in cell-to-cell synchronization, and the extent of their similarity to VGLUT2 waves varied distinctly. The portions of waves in VGLUT2 neurons were usually larger than those in GABAergic neurons (Fig. 3d). Notably, the waves in PV neurons were more similar to those in VGLUT2 neurons, while the waves in SOM neurons were less similar, with VIP neurons falling between the two (Fig. 3e). Most waves in the four types of neurons occupied similar frequency bands and demonstrated comparable energy concentrations within the frequency domain. However, some waves in GABAergic neurons, particularly those in SOM neurons, exhibited slightly greater frequencies than those in VGLUT2 neurons, and the frequency deviation of waves in GABAergic neurons often surpassed that of VGLUT2 neurons (Fig. 3f).

In addition, the spatial distributions of $FC$ and $FC_{GSR}$ were consistent across various cell types and approximately mirrored the spatial distributions of waves $\Phi_1$ and $\Phi_2$, as shown in Fig. 3h. Correlation matrices for waves $\Phi_1$ and $\Phi_2$ in these four cell types confirmed their

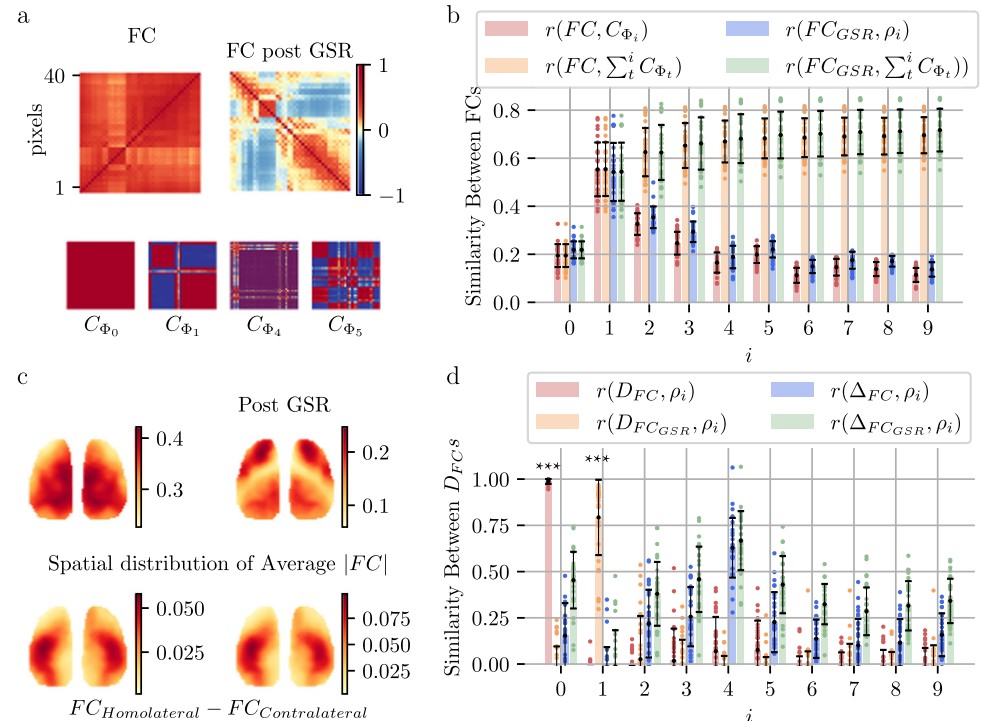

**Fig. 2 | FC as a linear superposition of zero-lag and time-lag synchronization structures. a** FC matrices were calculated for the original fluorescent signals ($FC$), GSR-regressed signals ($FC_{GSR}$), and time courses reconstructed from $\Phi_0$ ($C_{\Phi_0}$), $\Phi_1$ ($C_{\Phi_1}$), $\Phi_4$ ($C_{\Phi_4}$) and $\Phi_5$ ($C_{\Phi_5}$). While the values differ considerably, the $FC$ matrix and $FC_{GSR}$ matrix display a remarkable correlation ($r = 0.93 \pm 0.03$), suggesting a shared underlying trend in their relative positioning. **b** Similarities of the $FC$ and $FC_{GSR}$ maps with $C_{\Phi_i}$, including individual waves and their linear superpositions ($\sum_{t=0}^{i} w_t C_{\Phi_t}$), measured by the Pearson correlation coefficient. The error bars represent the mean ± standard error across the experiments. 23 experiments over $n = 10$ male mice. Source data are provided as a Source Data file. **c** Spatial distributions of pixel-averaged $FC$($D_{FC}$) and $|FC_{GSR}|$ ($D_{FC_{GSR}}$) with their

identities as $\Phi_{SM}$ and $\Phi_{SM2}$, respectively (low MSEs to correlation matrices of $\Phi_1$ and $\Phi_2$ in VGLUT2 neurons, Supplementary Fig. 3). The presence of similar waves at comparable frequencies across diverse neuronal types underscores their collaborative interactions and synchronizations.

### Distinct developmental trajectories of the synchronization structures in four neuron types

Although the FC of the four neuronal types converged in similar patterns, their developmental trajectories are quite different. SOM and VGLUT2 neurons establish complete FC before P14 and undergo only minor pruning between P14 and P56. PV neurons gradually establish FC between P14 and P56, while VIP neurons establish FC before P14 and between P28 and P56, with minor pruning between P14 and P28. Figure 4a shows the seed-seed FC and changes during development. The differences in FC between P14 and P56 are shown in Supplementary Fig. 2c.

The developmental trajectories of FC also exhibited regional specificity. Figure 4b illustrates the spatial distribution of the changes in RSFC strength in the cortex and reveals significant differences between GABAergic and VGLUT2 neurons. For GABAergic neurons, the increase in FC strength showed a pattern similar to the spatial distribution of $\Phi_1$ and was mainly concentrated in two regions. In contrast, VGLUT2 neurons primarily undergo pruning in different regions, resulting in more varied FC changes.

hemispheric differences ($\Delta_{FC}$, $\Delta_{FC_{GSR}}$). The hemispheric differences were calculated by the pixel average of $FC_{Homolateral} - FC_{Contralateral}$. **d** Similarities between the spatial distribution of FC ($D_{FC}$, $D_{FC_{GSR}}$) and the intensity distributions of standing/traveling waves (10 strongest waves, $\rho_i$, as shown in Fig. 1c), as measured by the Pearson correlation coefficient; only positive correlations are shown. $D_{FC}$ and $\rho_0$, $D_{FC_{GSR}}$ and $\rho_1$, $\Delta_{FC}$ exhibited significantly greater pairwise similarities than those of all the other combinations (Fisher's z-transformed two-sided $t$-tests, $p = 10^{-9}$, $p = 6.88 \times 10^{-6}$ after FDR correction with a threshold of 0.05). The error bars represent the mean ± standard error across the experiments. 23 experiments over $n = 10$ male mice. Source data are provided as a Source Data file.

Figure 4c, d also shows the differences in FC between GABAergic and VGLUT2 neurons, as well as their spatial diversity. During development, the average FC strength in GABAergic neurons usually gradually increases, while the average FC strength in VGLUT2 neurons displays more diverse patterns, including continuous increases, minor changes, and decreases following an initial increase. In terms of areas of significant connection, for SOM and VIP neurons, there is generally a gradual increase during development, but in a few cases, there is an initial decrease followed by an increase. The areas in the PV and VGLUT2 neurons often exhibited opposite trends at the same locations. In PV neurons, FC typically undergoes a decrease followed by an increase, while in VGLUT2 neurons, it undergoes an initial increase followed by a decrease.

In addition, GABAergic and VGLUT2 neurons exhibited differences in changes in short-range FC during development, with the former increasing and the latter decreasing slightly. This change is heterogeneous across cortical regions and varies between different neurons, coinciding with the establishment of their respective functional connectivity.

### Deviation of standing/traveling waves in different cells during development

Although FC was similar between neurons at P56, there was large variability in waves (Fig. 3), indicating that standing/traveling waves may capture more comprehensive information about brain synchro-

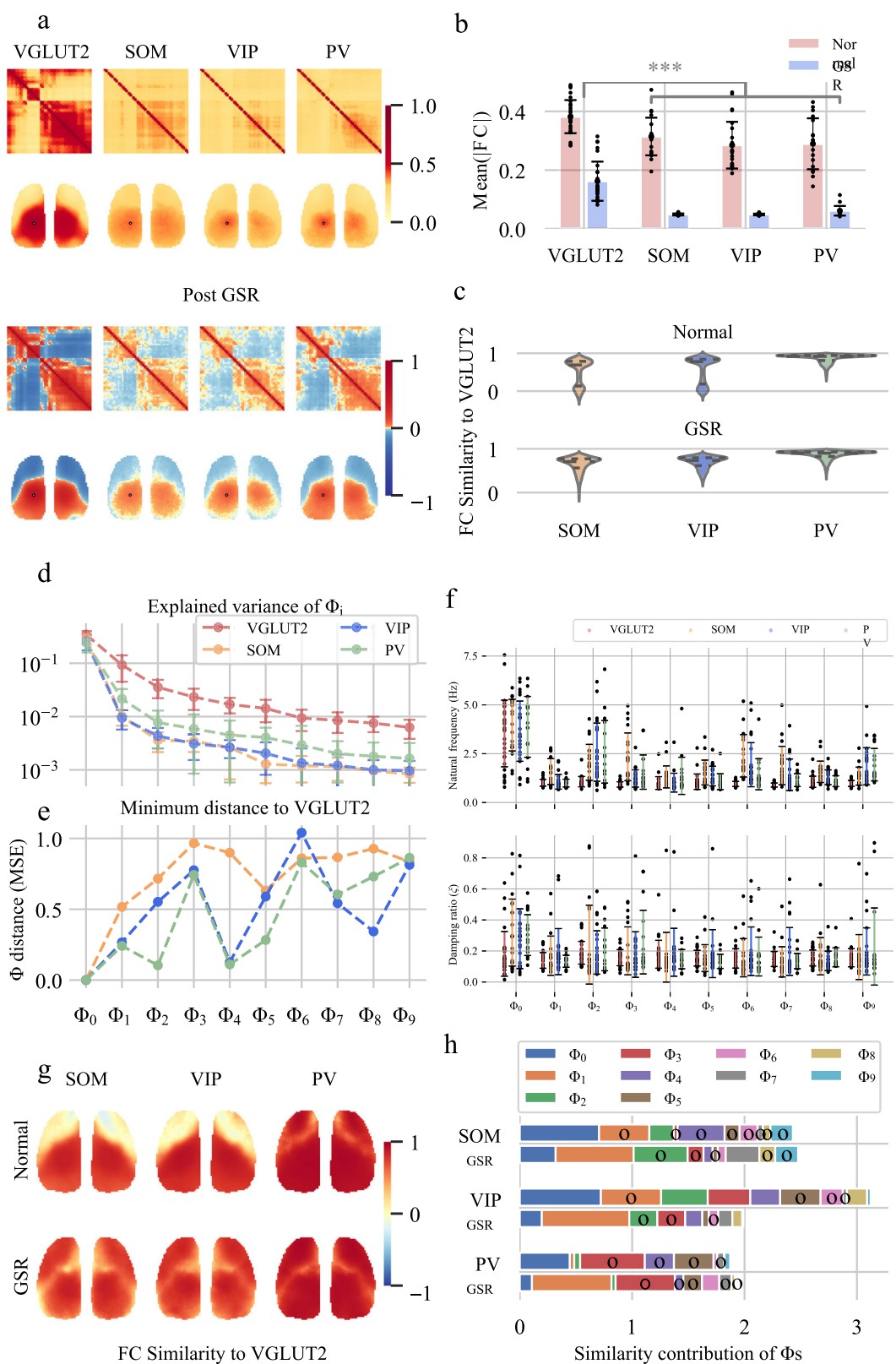

nization. We identified several typical waves across different neurons and developmental stages using the MSEs between the correlation matrix of waves in target conditions and that of VGLUT2 at P56 (Supplementary Fig. 3). Figure 5a, b, and c shows the deviations of three such waves, namely, $\Phi_{SM2}$, $\Phi_T$ and $\Phi_{DMN-like}$, respectively. The proportions of these components also vary with cell type and age. For VGLUT2 and SOM neurons, the proportion of each component usually

peaked at P28. PV neurons usually exhibited the opposite pattern to VGLUT2 neurons. The change in the proportion of VIP waves was consistent with the change in FC, with a significant increase after P28. For $\Phi_{DMN-like}$, in P56, it was more common in the PV and VGLUT2 neurons and rare in the SOM and VIP neurons.

The "content" of waves is clearly different in different cells and at different ages. As illustrated in Fig. 5d, certain waves in SOM and PV

**Fig. 3 | The four types of neurons exhibit similar FC patterns at P56. a** *FC* matrix and *FC*<sub>GSR</sub> matrix and *FC* and *FC*<sub>GSR</sub> maps from seed point *SSP-tr 3* of four types of neurons (VGLUT2, SOM, VIP, and PV). **b** The average of *FC* and |*FC*<sub>GSR</sub>| in the VGLUT2 neurons are significantly larger than those in the GABAergic neurons (Fisher's z-transformed two-sided *t*-tests, $FC : p = 4.8 \times 10^{-4}$, $FC_{GSR} : p = 5.3 \times 10^{-7}$, after FDR correction with a threshold of 0.05). The error bars represent the mean ± standard error across the experiments. (23 experiments over $n = 10$ VGLUT2 male mice. 17 experiments over $n = 13$ PV male mice. 20 experiments over $n = 10$ SOM male mice. 24 experiments over $n = 10$ VIP male mice.) **c** Violin plot of similarities between the FC of GABAergic neurons and that of VGLUT2 neurons at P56, measured by the Pearson correlation coefficient $S_i = Pearson\left(\left[C_{VGLUT2}(i,j) \, for \, j = 1,2 \ldots\right], \left[C_{GABAergic}(i,j) \, for \, j = 1,2, \ldots\right]\right)$, where $C(i,j)$ is the Pearson correlation coefficient between the signals of pixels $i$ and $j$. The similarities during development are shown in Supplementary Fig. 2b. **d** Variance explained by the top 10 waves in GABAergic and VGLUT2 neurons at P56. The error bars represent the mean ± standard error across the experiments. (23 experiments over $n = 10$ VGLUT2 male

mice. 17 experiments over $n = 13$ PV male mice. 20 experiments over $n = 10$ SOM male mice. 24 experiments over $n = 10$ VIP male mice.) **e** Degree of similarity between waves from VGLUT2 and GABAergic neurons at P56, quantified by the minimum mean squared error (MSE) of the correlation matrices from a specific wave to all waves in VGLUT2. A lower MSE indicates greater similarity. **f** Natural frequencies ($f$) and damping ratios ($\xi$) of the 10 strongest waves from four types of neurons. The error bars represent the mean ± standard error across the experiments. (23 experiments over $n = 10$ VGLUT2 male mice. 17 experiments over $n = 13$ PV male mice. 20 experiments over $n = 10$ SOM male mice. 24 experiments over $n = 10$ VIP male mice.) Source data are provided as a Source Data file. **g** The spatial distribution of FC similarities between VGLUT2 and GABAergic neurons at P56 was measured by the Pearson correlation coefficient, as shown in Fig. 3c. The spatial distributions of FC similarities during development are shown in Supplementary Fig. 2a. **h** The similarity between FC similarity to VGLUT2 and spatial distributions ($\rho_i$) of waves measured by Pearson correlation coefficients. The black circles denote negative correlations.

neurons exhibited a lower signal-to-noise ratio and lack a discernible spatial distribution pattern. These waves account for a relatively small proportion of the overall waves, indicating that they may be less significant in terms of their contribution to neural activity. These observations indicate that there are fewer waves in these two types of neurons than in VGLUT2 neurons. However, the CPCA method is unable to determine the number of waves. Therefore, we indicate the "quality" of waves by calculating the peak signal-to-noise ratio (PSNR) between the spatial distribution and its median filter (see Methods). As shown in Fig. 5e, VGLUT2 had the richest neural activity, and the PSNRs were the highest among the four types of neurons. PSNRs are typically smaller and decrease quickly as the wave number increases, which indicates that the information content decreases as the proportion of waves decreases. There are also differences in developmental stages, with minimal alterations in the content of waves in VGLUT2 and SOM neurons during development but a significant increase in that in PV neurons after P14 and in VIP neurons after P28. This phenomenon is also observed during the development of FC.

The frequency of waves in the PV, SOM and VIP neurons of young mice was greater than that in the VGLUT2 neurons of adult mice, and the waves were relatively chaotic, varying between the different experiments (Fig. 3f, Fig. 5f, g). VGLUT2 neurons are relatively more consistent with those of adults. This finding suggested that the frequency response characteristics of GABAergic neurons adjust during development to match those of excitatory neurons.

We analyzed neural activity in mice during movement when the mice were spontaneously running on a rotating plate. If we continue to apply the assumptions of CPCA, which states that each brain function corresponds to a specific waveform, it is interesting to decompose waves associated with behaviors such as running. Figure 6a shows that two components ($\Phi_G$ and $\Phi_{SM}$, identified by the MSEs between the correlation matrix of these waves and those in VGLUT2 neurons at P56; Supplementary Fig. 3) were decomposed during movement. During movement, several components were similar to those observed at rest (Supplementary Fig. 1), but the waves were severely deformed. As shown in Fig. 6c, the correlation matrix of $\Phi_4$ during movement appeared to be a combination of $\Phi_T$ and $\Phi_{DMN-like}$ at rest (Fig. 2), with MSEs of 0.756 and 0.831, respectively (Supplementary Fig. 3), which suggested that the lack of correlation between these waves disappeared. The changes in the waves in the four types of neurons during movement differed, and deterioration of "quality" was a common phenomenon. PV and VGLUT2 neurons were the most affected, SOM neurons were the least affected, and VIP neurons were between the two (Fig. 6b). This difference may be due to differences in the roles of different cells during spontaneous movement.

In the anesthetized state, the spatial distributions and characteristics of waves undergo noticeable changes. $\Phi_G$, $\Phi_{SM}$, $\Phi_{SM2}$, $\Phi_T$ and $\Phi_{DMN-like}$ were identified using the MSEs between the correlation

matrix of waves under target conditions and those in VGLUT2 neurons at P56 (Supplementary Fig. 3). As depicted in Fig. 7a, b, and c, there was a significant reduction in both the proportions and qualities of the waves with an increase in the burst suppression ratio (BSR). The BSR serves as a measure of anesthesia depth, offering an advantage over the anesthetic concentration by reducing the impact of individual animal variations. Notably, the proportion of $\Phi_{DMN-like}$ waves gradually increased with stronger anesthesia (increased BSR). During anesthesia, the frequency characteristics of the waves were greater than those in the awake state, and the waves exhibited a relatively chaotic nature, varying between different experiments, as shown in Fig. 8c-e.

Coactivation patterns (CAPs) under anesthesia are predominantly characterized by alternating activations between central and peripheral cortical regions, as illustrated in Fig. 8b. Correspondingly, the standing wave $\Phi_G$ under anesthesia (Fig. 8a) shows not only overall changes but also detailed variations. Notably, a wave propagated from the retrosplenial area (RSP) to the lateral cortex or vice versa under anesthesia that varies with different anesthetics, as shown in the Supplementary Movie 1. The spatial pattern of $\Phi_G$ closely resembled the main CAPs. Moreover, the global signal also contained traveling components that were not addressed by GSR, potentially highlighting a limitation of GSR.

## Gene expression is associated with traveling waves

To understand the molecular mechanisms underlying the structure of large-scale neural activity, we compared the spatial patterns of the traveling waves with the expression patterns of 215 specific genes (Fig. 9). Notably, we found that the spatial distribution patterns of cortical gene expression were similar to those of traveling waves, suggesting that gene expression might be the basis for the formation of these waves.

Our results showed that the spatial distribution of $\Phi_{SM}$ was strongly correlated with 186 genes, with an absolute correlation coefficient greater than 0.3. The genes with the strongest correlation were *Wdr5* (r = 0.76) and *E2f6* (r = − 0.85). Interestingly, $\Phi_{SM}$ was the most prominent component after GSR and was also correlated with the expression of most genes analyzed in our study. We also found that spatial patterns of gene expression positively or negatively correlated with $\Phi_{SM}$ were usually located in different gene coexpression trees.

Although the number of genes related to the other three waves was low, these genes still hold potential physiological significance. The numbers of genes (r > 0.3) correlated with $\phi_{SM2}$, $\phi_T$, and $\phi_{DMN-like}$ were 10, 11, and 16, respectively, and the genes with the highest correlation with each wave were *Neto2*, *Stk32c*, and *Man1a*, respectively. We analyzed genes correlated with these three waves (r > 0.3) and genes correlated with the $\Phi_{SM}$ (r > 0.75, 16 genes), as shown in Fig. 9c. Our results showed that these genes can be roughly divided into two categories. The genes in the first category are related to cell

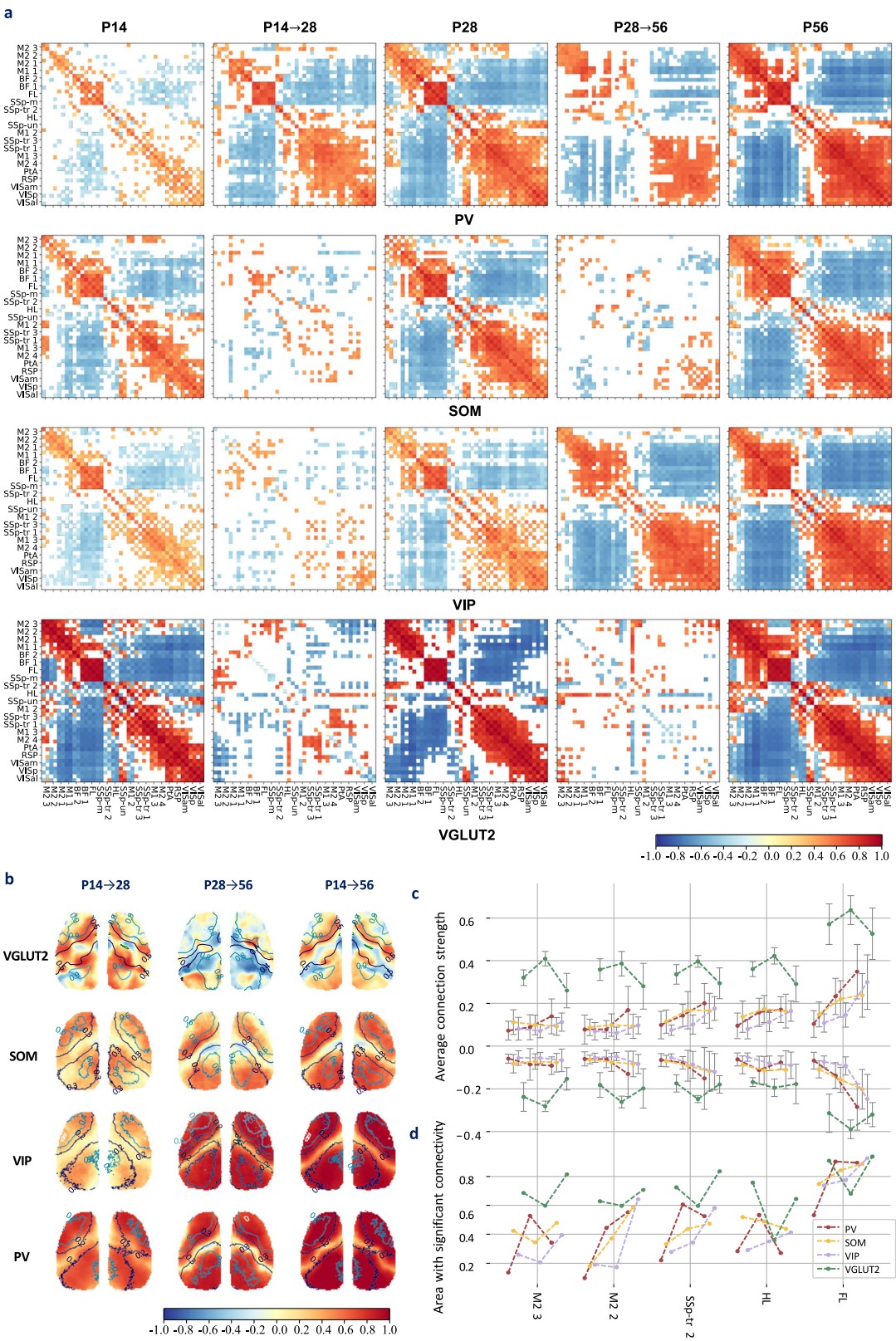

differentiation, cellular component organization, and the establishment of localization; these genes may be involved in the development of structures in the brain, which may be related to the structural basis of FC and traveling waves. The genes in the second category are related to the response to stimuli, protein metabolic processes and signaling, which may be related to the interaction of information in the neural network and crucial for proper functioning of FC and traveling waves.

## Discussion

Complex global synchronization structures in various cell types, ages, and physiological states. These structures, present in the spontaneous activity of the brain, show a certain level of conservation across different cell types, ages, and even species (humans and mice)[24,43]. Although state-dependent FC has been identified in various contexts, the synchronization of global neural activities extends beyond the

**Fig. 4 | The developmental processes of FC are cell type and region specific.**
**a** Significant seed-seed FC post-GSR and changes in four types of neurons across three developmental stages, with diagonal entries representing the short-range FC and changes ($p < 0.05$, Fisher's z-transformed two-sided $t$-tests, after FDR correction with a threshold of 0.05). Changes between P14 and P56 are shown in Supplementary Fig. 2a. **b** The spatial distribution of changes in the average FC post-GSR during development, measured by the difference in pixel averages of correlation coefficients. The contour shows the spatial distribution of $\Phi_1$. **c** Trends in the average FC at 5 seed points during development. The positive connections and negative connections are averaged separately and are shown in the top and bottom

panels, respectively. The error bars represent the mean ± standard error across the experiments. (P14: 13 experiments over $n = 6$ VGLUT2 male mice. 17 experiments over $n = 6$ PV male mice. 16 experiments over $n = 16$ SOM male mice. 23 experiments over $n = 5$ VIP male mice. P28: 8 experiments over $n = 4$ VGLUT2 male mice. 32 experiments over $n = 7$ PV male mice. 19 experiments over $n = 6$ SOM male mice. 17 experiments over $n = 7$ VIP male mice. P56: 23 experiments over $n = 10$ VGLUT2 male mice. 17 experiments over $n = 13$ PV male mice. 20 experiments over $n = 10$ SOM male mice. 24 experiments over $n = 10$ VIP male mice.) **d** Area of regions with significant connections at 5 seed points.

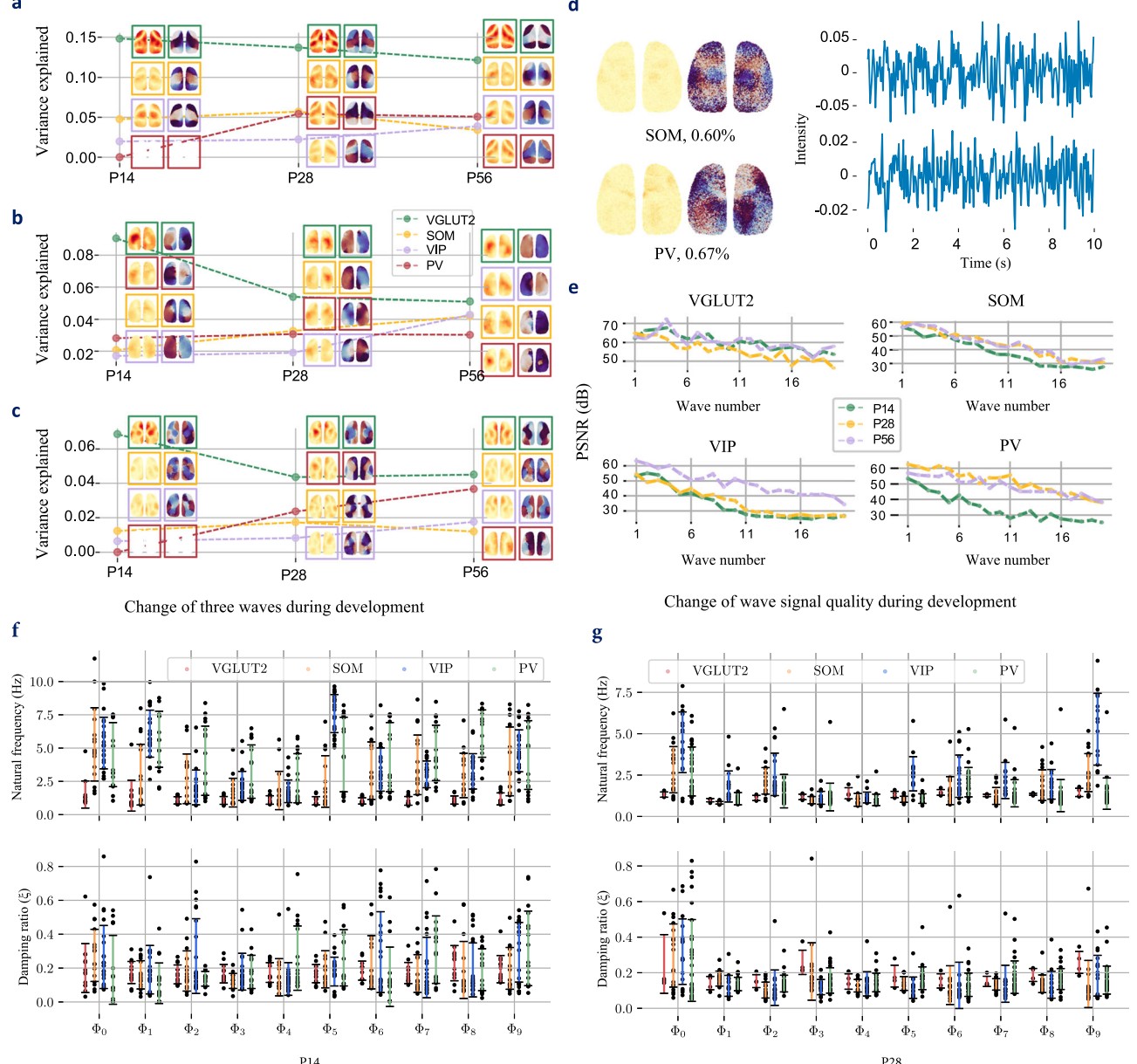

**Fig. 5 | Spatial variations in waves during development. a–c** Changes in $\Phi_{SM2}$, $\Phi_T$ and $\Phi_{DMN-like}$ in the four neuron types at three developmental stages (relative to post-GSR). The dashed lines show the changes in the corresponding wave proportions during development, and the images show the changes in the corresponding wave spatial distributions. **d** Spatial distributions and waveforms (first 30 s in 180 s acquisition) of "low-quality" waves. **e** Changes in wave quality during development, measured by the PSNR between the spatial distribution and its median filtering. The qualities of the 20 strongest waves are presented. **f-g** Natural frequencies ($f$) and damping ratios ($\xi$) of the 10 strongest waves from four types of

neurons at P14 and P28. The error bars represent the mean ± standard error across the experiments. (P14: 13 experiments over $n = 6$ VGLUT2 male mice. 17 experiments over $n = 6$ PV male mice. 16 experiments over $n = 16$ SOM male mice. 23 experiments over $n = 5$ VIP male mice. P28: 8 experiments over $n = 4$ VGLUT2 male mice. 32 experiments over $n = 7$ PV male mice. 19 experiments over $n = 6$ SOM male mice. 17 experiments over $n = 7$ VIP male mice. P56: 23 experiments over $n = 10$ VGLUT2 male mice. 17 experiments over $n = 13$ PV male mice. 20 experiments over $n = 10$ SOM male mice. 24 experiments over $n = 10$ VIP male mice.) Source data are provided as a Source Data file.

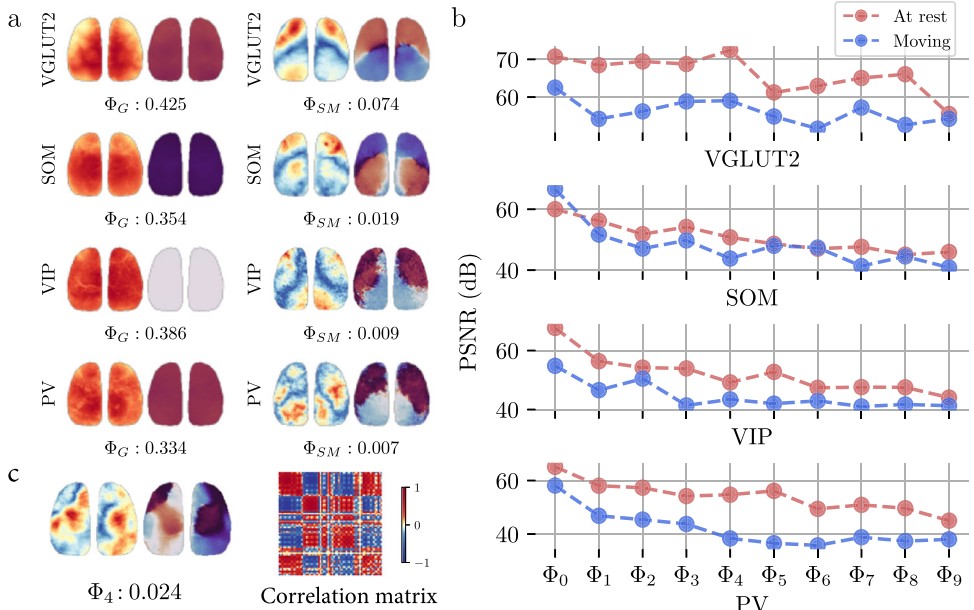

**Fig. 6 | Waves during movement extracted by CPCA. a** Spatial distributions of the waves $\Phi_G$ and $\Phi_{SM}$ during movement (running on plate) in four types of neurons. **b** Changes in wave quality during movement, measured by the PSNR between the spatial distribution and its median filtering. The qualities of the 10 strongest waves are presented. **c** Spatial distribution and correlation matrix of $\Phi_4$ during movement.

simple superposition of synchronized or desynchronized states[8–14]. FC encompasses a more intricate picture as a highly multidimensional continuum of synchronizations, including a time lag (or traveling waves, which may be initially interpreted as a desynchronized state) and differences between cells[8].

While the synchronized structures of the three GABAergic neurons exhibited similarities to those of VGLUT2 neurons, the traveling waves revealed marked differences in cell-to-cell synchronization. The glutamatergic neurons exhibited greater proportions and greater diversity of wave patterns than did the GABAergic neurons. Among the three GABAergic neuron types, the waves observed in PV neurons exhibited the closest resemblance to those in VGLUT2 neurons, while those in SOM neurons exhibited the least resemblance. The proportion of $\Phi_{DMN-like}$ waves in PV and VGLUT2 neurons was greater, indicating a potential role of PV in DMN function, consistent with previous studies[44].

A negative correlation is a characteristic of time-lag synchronization. Global signal regression (GSR) is a controversial method commonly employed in fMRI studies[45]. Previous studies have suggested that global signal regression (GSR) can introduce a more pronounced negative correlation in FC[46]. Compared to fMRI and ultrafast fMRI[40], our method offers finer spatiotemporal resolution and enables almost direct imaging of intracellular signals, unaffected by HRF, potentially capturing more characteristics of neural activity. Using fluorescent signals, we investigated the impact of GSR on FC. Our analysis revealed that while GSR eliminates zero-lag synchronization, it preserves FC structures, which reflect time-lag synchronization. Our analysis revealed negative correlations in spontaneous neural activity arising from the propagation of specific wave patterns. These interferences can lead to near-zero correlation coefficients between activity in different brain regions, some of which may be interpreted as desynchronization. These findings address a gap in understanding resting-state FC (rsFC) data, particularly in human fMRI studies.

A complex picture of brain synchronization under anesthesia. Anesthesia can induce the brain to enter a more synchronized state, a slow-wave state similar to sleep[8]. In this state, pancortical synchronized activity is dominant, as evidenced by our results for $\Phi_0$. However, we observed the presence of traveling waves other than these

slow oscillations, suggesting a more complex synchronization structure, which is consistent with the findings of previous studies[8]. Furthermore, although all slow oscillations (all $\Phi_0$ under anesthesia) exhibit almost zero-lag synchronized activity, slow waves induced by different anesthetics display distinct characteristics in terms of spatial movement on a small scale. As anesthetics typically affect the GABAergic system[47], this might be a comprehensive outcome related to the mechanisms of synchronization, including interactions between GABAergic and glutamatergic neurons. This phenomenon warrants further investigation.

Cell-to-cell differences in movement impact spontaneous activity, although the main components are consistent with those at rest. We found that, compared with FC, traveling waves capture more detail about changes in neural activity and are consistent with subtype specificity found in previous studies[48]. All four types of neurons exhibited a decrease in wave quality; PV and VGLUT2 neurons were the most affected, SOM neurons were the least affected, and VIP neurons were between the two. Previous studies have shown that PV neurons split into distinct populations during movement: an excitatory group and a suppressed group[49]. This functional dichotomy likely explains the dramatic decrease in wave quality we observed in PV neurons.

Different developmental trajectories in FC and traveling waves are observed for GABAergic interneurons and glutamatergic neurons. FCs of SOM and glutamatergic neurons are established at an early stage, while the FC of PV neurons establishes connections throughout mouse development, and rapid expression of PV and establishment of synapses are among the most critical changes in the GABAergic system from P14–P56[50]. The FC of VIP neurons is established after adolescence. Despite VIP expression being abundant during embryonic and early life stages[51], it has not been determined why the FC of these neurons is mainly established between P28 and P56. VGLUT2 and SOM neurons establish relatively complete synchronizing structures at early stages. While VGLUT2 neurons exhibit nearly perfect time-lag synchronization even at early stages, GABAergic neurons, except SOM neurons, lack this type of synchronization. Studies indicate that neural desynchronization may stem from shared input/output among excitatory and inhibitory neurons[1,2]. The robustness of traveling waves in VGLUT2 neurons at an early age presents a challenge for determining

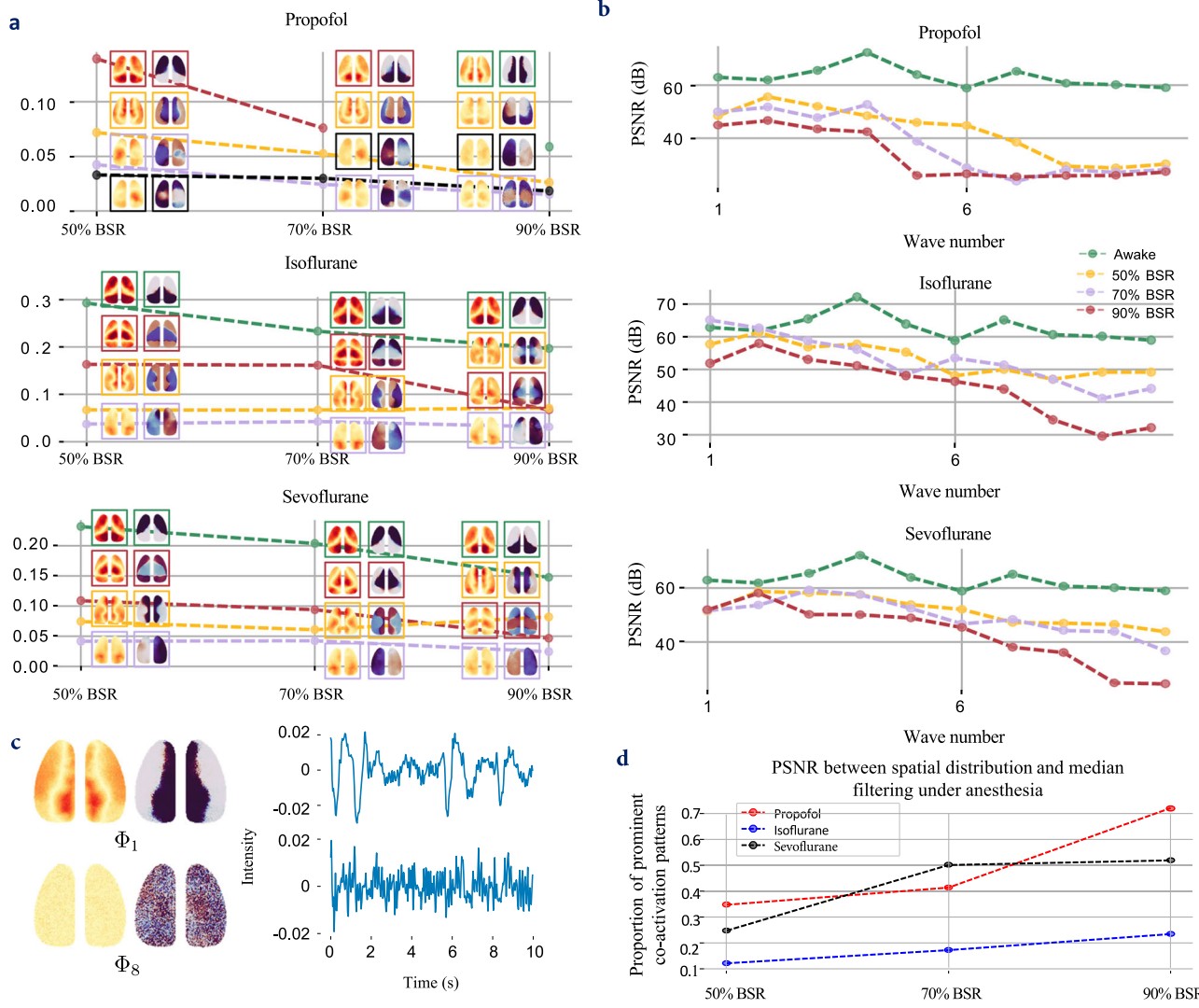

**Fig. 7 | Traveling waves in VGLUT2 neurons vary under anesthesia. a** Changes in the $\Phi_{SM}$, $\Phi_{SM2}$, $\Phi_T$ and $\Phi_{DMN-like}$ waves of four cell types (post-GSR, relative) under anesthesia are illustrated, with different waves represented by different colors. The dashed lines show how the proportions of these waves change as the burst suppression ratio (BSR) increases, while the images depict the corresponding changes in their spatial distributions. **b** Changes in wave quality under anesthesia, measured by the PSNR between the spatial distribution and median filtering. $\Phi_{SM}$ disappeared under Propofol anesthesia under 90% BSR (red line). **c** Spatial distributions and waveforms (first 30 s in 180 s acquisition) of $\Phi_1$ and $\Phi_8$ (propofol, 90% BSR). $\Phi_8$ is a low-quality wave. **d** Proportions of prominent coactivation patterns (CAPs) across different BSRs.

the mechanisms underlying time-lag synchronization due to the immaturity of GABAergic neurons.

The synchronization of spontaneous activity is consistent with the changes in gene expression levels[52]. Gene transcripts have been linked to resting-state FC and networks in fMRI studies[53–57], which is consistent with our findings. Our approach, which combines gene analysis with spontaneous activity, holds great promise for identifying the functional roles of resting-state activities. However, how this similarity arises remains unclear. By investigating the correlation between genes and traveling waves, we can identify genes that warrant further investigation and subsequently delve into their functions. This approach may help to decipher the role of synchronizations in the brain. For example, the expression pattern of the gene *Epha6* is similar to the spatial distribution of time-lag synchronization $\Phi_5$, and *Epha6* is related to memory function[58], which suggests that $\Phi_5$ may play a role in memory formation. Since brain synchronizations exhibit similarities across different species[43], the insights gained from this method in mice could be extrapolated to other species. Thus, this methodology has

significant potential for elucidating the relationships between genetic pathways and the interplay between brain structure and function.

Interactions between different types of neurons. We observed opposite developmental trajectories between PV and VGLUT2 neurons in terms of traveling waves. PV expression and rapid synapse formation occur during development[50] and PV inhibits glutamatergic neurons directly[48]. The changes in VGLUT2 traveling waves from P14 to P56 may be associated with the establishment of PV synchronization. PV neurons inhibit the cell bodies and proximal dendrites of pyramidal cells, while SOM neurons target distal dendrites[59–61]. Horizontal propagation in layer 5 pyramidal cells during the up state mainly occurs among their cell bodies[62]; thus, inhibition by PV neurons at the cell bodies is likely the primary inhibitory system, with SOM neurons inhibition of distal dendrites acting as a secondary system[62]. However, the impact of cell type on time-lag synchronization has not been determined. Notably, the timing of the maturation of synchronization structures differs across cell types. After P28, VIP synchronization matures rapidly, while VGLUT2 synchronization exhibits a pattern seemingly opposite to that

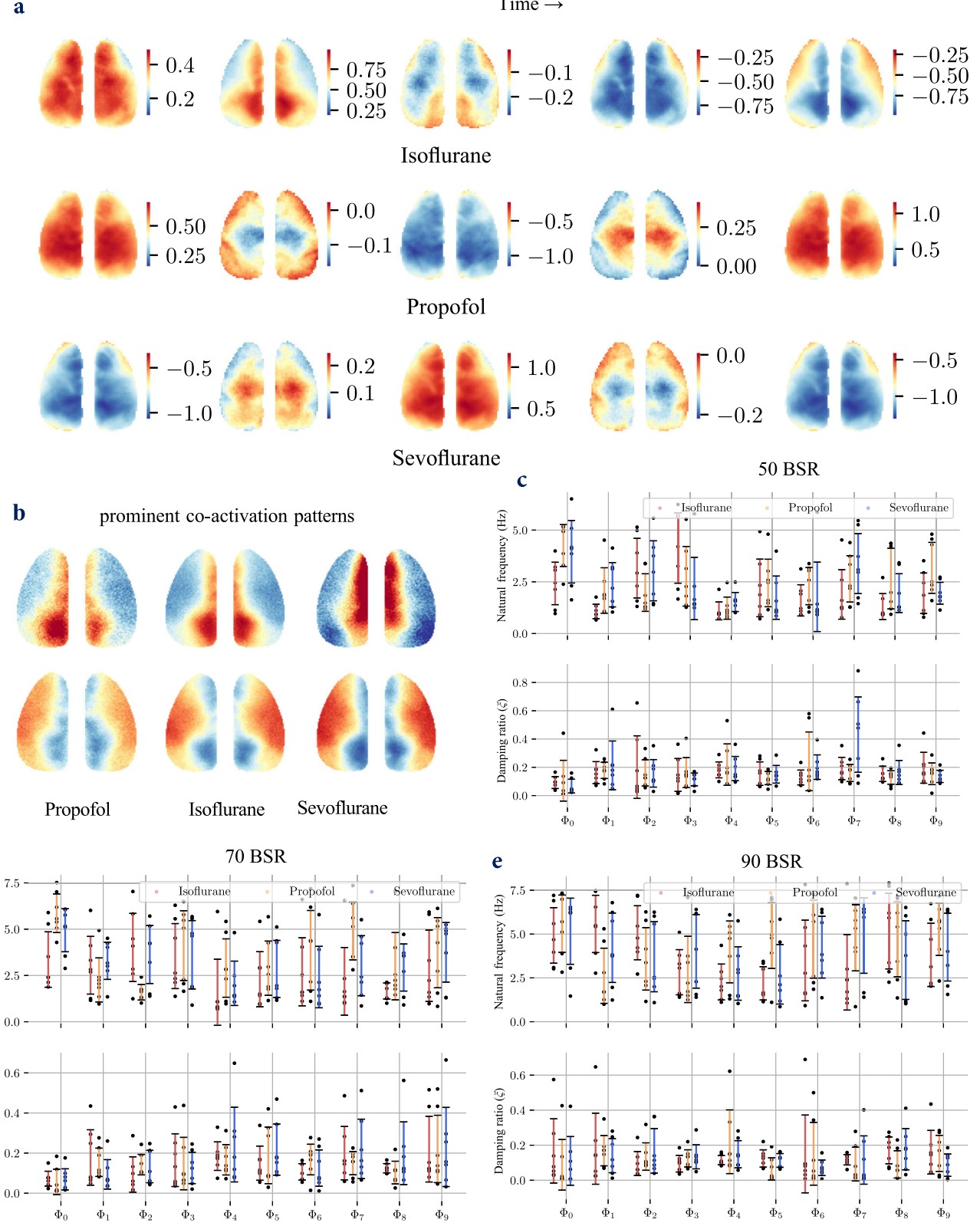

**Fig. 8 | Standing waves and frequency characteristics of VGLUT2 neurons under anesthesia. a** Spatiotemporal patterns of the $\Phi_G$ wave under anesthesia at a BSR of 90%. **b** Prominent coactivation patterns in the anesthetized state (90% BSR). **c**–**e** Natural frequencies ($f$) and damping ratios ($\xi$) of the 10 strongest waves from VGLUT2 neurons at BSR levels of 50%, 70% and 90%. The error bars represent the mean ± standard error across the experiments. $n = 7$ male mice examined over 9 independent experiments; Source data are provided as a Source Data file.

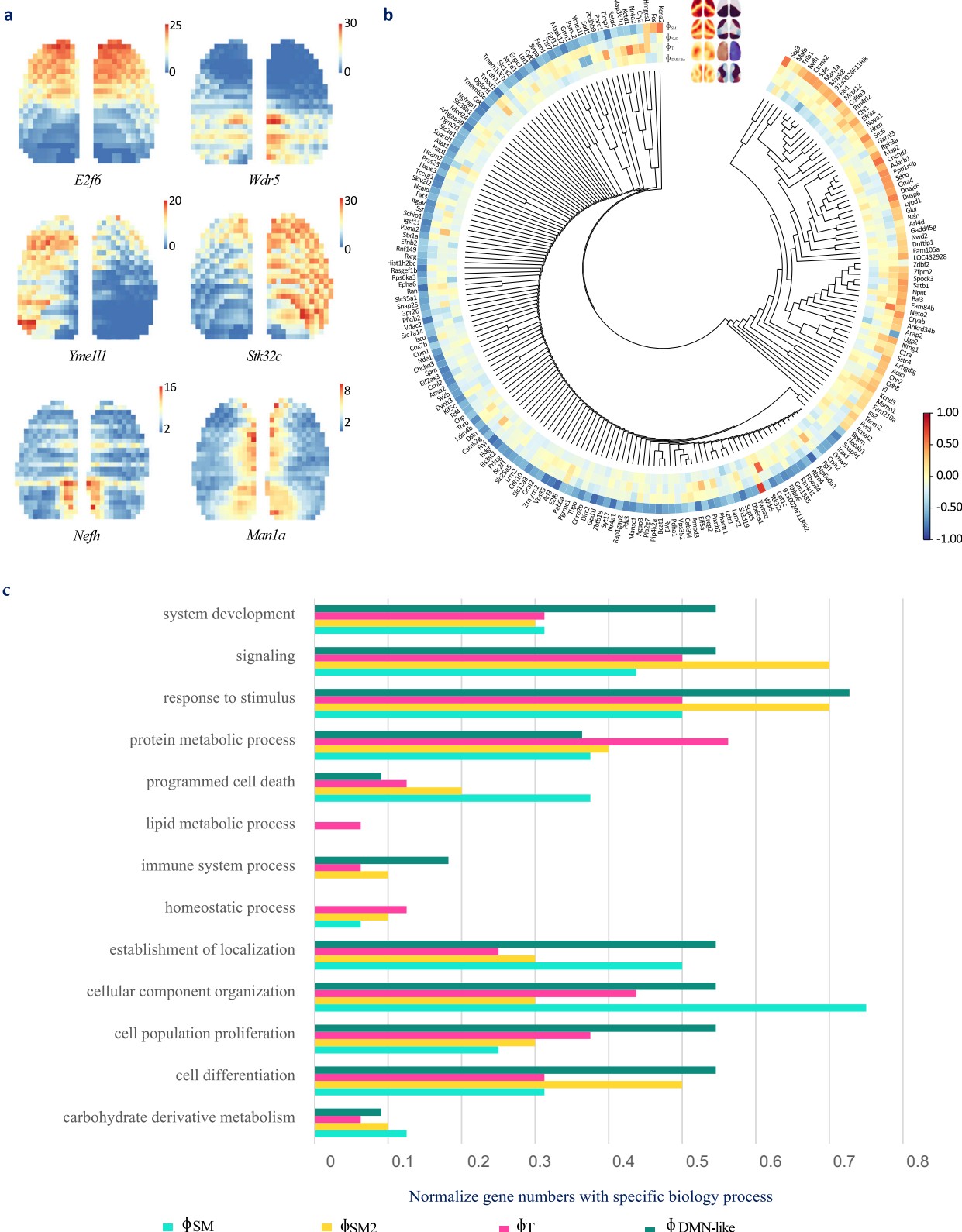

**Fig. 9 | Gene expression associated with traveling waves. a** Spatial distribution of gene expression in the cortex that shows patterns similar to those of waves. The colors indicate the intensity of gene expression on the cortex surface. **b** Similarity between the spatial distribution of gene expression and the spatial distribution of the $\Phi_{SM}$, $\Phi_{SM2}$, $\Phi_{T}$ and $\Phi_{DMN-like}$ waves illustrated by a gene coexpression tree, measured by the correlation coefficient. The colors in the heatmap show the correlation coefficients. Source data are provided as a Source Data file. **c** Biological processes associated with genes that are correlated with traveling waves.

observed before P28. Before P28, the decrease in VGLUT2-related traveling waves coincided with the increase in these waves in PV neurons. However, after P28, the traveling waves of VGLUT2 align with those of both PV and VIP neurons, coinciding with modifications in the presynaptic terminal during this period[63] and resulting in a more complex picture. SOM neurons exhibit early development of stable synchronizing structures. In neural systems, excitatory activity typically triggers balanced inhibition for stability[64]. VGLUT2 neurons rapidly mature during the early stages of eye opening[65], and SOM neurons form synaptic connections with excitatory neurons before P14[66], possibly leading to early and relatively complete synchronizations similar to those observed in VGLUT2 neurons.

## Methods

Our experimental setup and data processing method are described in Supplement Fig. 4. We monitored cortical activity in four mouse lines using fluorescence imaging. Each pixel in the imaging results represents a time series of changes. To obtain an FC map for each time series, we calculated its correlation with the other time series and subsequently clustered these FC maps to aid in the selection of seed points.

### Animals

We bred RCL-GCaMP6s mice[67] (Ai96; B6.129S6-Gt(ROSA) 26Sor^tm96(CAG-GCaMP6s)Hze/J, Jax #024106) with VGLUT2-IRES-Cre mice (STOCK Slc17a6^tm2(cre)Lowl/J, Jax #016963), Pvalb-IRES-Cre mice (B6;129P2-Pvalb^tm1(cre)Arbr/J, Jax #008069), SOM-IRES-Cre mice (STOCK Sst^tm2.1(cre)Zjh/J, Jax #013044) and VIP-IRES-Cre mice (STOCK VIP^tm1(cre)Zjh/J, Jax #010908), generating "VGLUT2-GCaMP6s", "PV-GCaMP6s", "SOM-GCaMP6s", and "VIP-GCaMP6s" mice, respectively[68]. We imaged 6 VGLUT2 mice, 6 PV mice, 6 SOM mice, and 5 VIP mice at P14. At P28, we imaged 4 VGLUT2 mice, 7 PV mice, 6 SOM mice, and 7 VIP mice. At P56, we imaged 10 VGLUT2 mice, 13 PV mice, 10 SOM mice, and 10 VIP mice. All mice were maintained on a standard 12 h light-dark cycle in a room with controlled temperature (22 ± 1 °C) and humidity (50 ± 5%) and had free access to food and water. All animal procedures were approved by the Hubei Provincial Animal Care and Use Committee and adhered to the experimental guidelines of the Animal Experimentation Ethics Committee of Huazhong University of Science and Technology in China.

### Surgical preparation

To enable repeated head-fixed imaging, a chronic through-bone window was fitted on each mouse, as previously described[69,70]. After the skull was exposed, clear-drying dental cement was applied to the intact skull, followed by fixation of an awake imaging fixator and a glass coverslip to create a partially transparent imaging surface ~8 mm in diameter located 3 mm anterior to the bregma. The clear-drying dental cement used was a mixture of C&B Metabond (Parkell, Edgewood, NY, USA), C&B Metabond powder (product: S399), C&B Metabond Quick Base (product: S398), and C&B Universal catalyst (product: S371). During surgery, the mouse was placed in a stereotactic apparatus (Riverward, Shenzhen, China) under inhalational anesthesia with isoflurane (2% for induction, 1.5% for maintenance), while body temperature was maintained at 37 ± 0.5 °C using a heating pad (RightTemp!, Kent Scientific, USA).

### Imaging

We fixed the brains of awake mice and imaged them in a calcium fluorescence imaging system (see Supplement Fig. 1) as previously described[71]. We used a frame rate of 30 Hz and a resolution of 512 × 512 with 4 × 4 binning. Illumination was provided by a high-power mercury lamp (UHGLGPS, Olympus; 130 W) through a liquid light guide (ULLG150/300, Olympus). The excitation beam, ~488 nm for GCaMP fluorescence, was created by an excitation filter (FF01-480/40-25, Semrock) and reflected onto the cortical surface with a dichroic mirror (FF495-Di03 25 × 36, Semrock). Video was captured using an sCMOS

camera (16 bits, 65 × 6.5 µm, Flash4.0V2C11440-22CU, Hamamatsu, Japan), with a dichroic mirror filtering out GCaMP6f excitation light. The objective lens was defocused down by ~400 µm to minimize vascular artifacts. Fluorescence imaging was performed in a darkened, sound-proof chamber after 15 min of acclimation. We monitored the mice via a body camera throughout the imaging process. The fluorescent signals showed obvious changes, as demonstrated in Supplementary Fig. 1. Under anesthesia, electroencephalogram (EEG) data were monitored using two stainless steel screws implanted into the prefrontal cortex.

### Preprocessing

We selected fluorescent calcium signals on the recordings from the body camera, excluding periods with obvious body movement. The signals were bandpass filtered from 0.1 to 14.5 Hz (Chebyshev type II digital filter). We registered the fluorescent calcium signals using both the Allen Mouse Common Coordinate Framework v2 (CCF v2) anatomical template[72] and the Allen Developing Mouse Brain Atlas[73] as references, as shown in Supplementary Fig. 4e. We selected data from mice with fluorescent signals that were significantly greater than those of wild-type mice (Supplementary Fig. 4g). The BSR under anesthesia was calculated by segmenting the EEG data into bursts and suppressions using voltage- and duration-based thresholds.

### FC

The strength of the connection was indicated by the Pearson correlation coefficient of the preprocessed fluorescent calcium signals with/without GSR. The correlation coefficients were subjected to Fisher Z transformation before being hypothetically tested. One-sample two-sided $t$-tests were used to evaluate the null hypothesis that the correlation means equal zero for all correlation coefficient groups. Additionally, Welch's unequal variances $t$-test was used to compare all pairwise correlations between two developmental timepoints. To account for multiple testing and control the false discovery rate (FDR), we corrected the $p$-values using the Benjamini–Hochberg algorithm[74]. We considered $p$-values <0.05 to indicate statistical significance.

The absolute average FC was determined by taking the average of the absolute values of all the $|FC_t^i|$ at each pixel. This process was executed by selecting each pixel $i$ and computing the FC map $FC_i^j, for\ j = 1, 2, \ldots$. The FC map illustrated the strength of the connections between the seed point and all other voxels in the cortex. The absolute average FC was then determined by averaging the absolute values of the FC map, $|\bar{FC}_t| = Ave(|FC_t^j|, j = 1, 2, \ldots)$. The hemispheric differences in FC were estimated by subtracting the average absolute FC of the ipsilateral hemisphere and $|\bar{FC}_p| = Ave(|FC_p^j|, j = ipsilateral\ pixels\ of\ p)$ from that of the contralateral hemisphere $|\bar{FC}_c| = Ave(|FC_c^j|, j = contralateral\ pixels\ of\ p)$ as follows: $(|\bar{FC}_p| - |\bar{FC}_c|)$. The FC similarity between pixels $i$ and $j$ was measured by Pearson's correlation coefficient between the FC maps, $Pearson([FC_i^t\ for\ t = 1, 2, \ldots], [FC_j^t, for\ t = 1, 2, \ldots])$.

The seed points were selected by clustering. The pixels were grouped into several classes by the k-means algorithm based on their connection maps. The number of means ranged from 2–25 and were selected according to the intragroup similarity and the distribution of groups in the cortex. We selected seed points based on the cluster results and CCF, attempting to ensure at least one point in each group. The connections between seed points were represented by Pearson's correlation coefficients. The short-range functional connectivity was calculated between the seed point and its adjacent region of 8 pixels. Figure 4f shows all the selected seed points.

### Gene expression

Our study employed in situ hybridization (ISH) data for 2,080 genes from the Allen Mouse Brain Atlas, which included 4,345 coronal

sections with masked resolution resampled to 200 microns[72]. To analyze the gene expression data, we employed weighted gene coexpression network analysis (WGCNA) and obtained distinct modules[38,53]. From these modules, we selected 214 genes based on the similarities between their spatial distribution of expression and the distribution of traveling waves. For each module, we visualized the gene expression data in a heatmap to identify genes that exhibited similar expression patterns to our coactivation pattern. We then compared the expression similarity of each gene with the spatial pattern of the traveling waves. The coefficient of correlation (r) was used to determine the similarity between the spatial distribution of gene expression patterns and the spatial patterns of traveling waves. Typically, a correlation coefficient (r) less than 0.3 is considered to be negligible[75], indicating a weak relationship between two variables[64]. On the other hand, a correlation value (r) above 0.75 is indicative of a high degree of correlation[76], suggesting a strong association between the spatial patterns of gene expression and traveling waves. Thus, we chose 0.3 and 0.75 as thresholds for analyzing the correlations between traveling waves and genes.

## Standing/traveling waves

We utilized complex principal component analysis (CPCA) to extract temporospatial waves from preprocessed fluorescent calcium signals. CPCA identifies zero-lag and time-lag spatial structures as several traveling and standing waves[15,37], and we examined 10 components for all obvious spatial structures. We also analyzed data collected under anesthesia and during movement and compared them to the results obtained using data at rest.

The CPCA method involves applying a Hilbert transform to the original signal, followed by principal component analysis (PCA). Our fluorescent signal $S_p(t)$ at the pixel ($p$) is a time ($t$) series. Each $S_p$ is first z scored ($N_p = \frac{S_p - Mean(S_p)}{\sqrt{Var(S_p)}}$) and then subjected to a Hilbert transform, $C_p = Hilbert(N_p)$. The outcome of the Hilbert transform is a complex number that forms a matrix:

$$M = \begin{pmatrix} C_{p_0}(t_0) & \cdots & C_{p_N}(t_0) \\ \vdots & \ddots & \vdots \\ C_{p_0}(t_M) & \cdots & C_{p_N}(t_M) \end{pmatrix} \qquad (1)$$

Principal component decomposition of $M$ yields scores ($t_k(t)$ for the k-th component, $l^2$-normalized as $\varphi_k(t)$), eigenvalues ($d_k$ for the k-th component) and loadings ($l_k(p)$ for the k-th component, $l^2$-normalized as $\rho_k e^{\theta_k}$). $l_k$ represents the spatial distribution of standing/traveling waves (complex numbers, including the amplitude $\rho_k$ and phase $e^{\theta_k}$ distributions). The real part of $t_k$ represents the waveforms of standing/traveling waves, while $d_k^2/\sqrt{n_{time} - 1}$ represents the variances of each wave. Combining scores and loadings or projecting the original signal with loadings yields the spatiotemporal pattern of each standing/traveling wave (Supplementary Movie 1).

For group-level analysis, we normalized the signals obtained from each experiment $N1_p, N2_p, \ldots$ and concatenated them over time to form a longer signal ($NG_p = [N1_p(t_0), \ldots, N1_p(t_n), N2_p(t_0), \ldots, N2_p(t_n), \ldots]$); then, the same steps were repeated as in a single experiment. The effectiveness of CPCA depends on the length of the signal; generally, the longer the signal is, the greater the signal-to-noise ratio.

There is no direct method for comparing different waves from CPCA results. To identify corresponding waves, we examined the connectivity matrices of the waves ($C_{\Phi_i}$). We started with the spatiotemporal pattern of each wave and computed its connectivity matrix, the correlation coefficients between pixels ($C_{\Phi_i}(j,k) = Pearson([S]_R^j(t),$

$t = 1, 2, \ldots, [S]_R^k(t), t = 1, 2, \ldots$), where $S_R^j(t)$ is the signal of pixel $j$ reconstructed from wave $\Phi_i$ at time $t$). We then compared these connectivity matrices based on the mean square error (MSE), considering the waves with the lowest MSE as being the same type. The mean square error is shown in Supplementary Fig. 3.

The quality of the traveling waves was measured by the PSNR between the spatial distribution and its median filtering. The median filtering had a kernel size of 10×10, and the real and imaginary parts were computed separately and then summed. The formula was as follows:

$$SNR_{total} = PSNR\left(Real\left(\rho e^\theta\right)\right) + PSNR\left(Imag\left(\rho e^\theta\right)\right) \qquad (2)$$

$$PSNR(x) = 10 \times \log_{10}\left(\frac{x_{\max}}{var\left(x - x_{filtered}\right)}\right) \qquad (3)$$

We normalized each pixel prior to conducting CPCA. This normalization allowed for comparability of signal magnitudes across different experiments and locations. However, the effectiveness of this method needs to be demonstrated. To investigate the differences in the spatial distribution and efficiency of fluorescent protein expression, we conducted simulation experiments (Supplementary Fig. 5). We generated three sets of unrelated signals and created three groups of spatial amplitude and time delay distributions, also adding zero-mean Gaussian noise. To mimic variations in fluorescent protein expression and spatial distribution, we produced several spatial distribution patterns (simulated efficiency in Supplementary Fig. 5). The generated signals were linearly superimposed and coupled with zero-mean Gaussian noise and simulated fluorescent protein expression spatial distributions to create simulated neural activity signals, which were subsequently subjected to CPCA. We conducted three sets of simulations with different sizes and spatial distributions of simulated efficiency. The results indicated that the differences in expression patterns or efficiencies of each marker did not significantly affect the outcomes of CPCA.

## Frequency

The power spectral density (PSD) of the fluorescence in Fig. 1d was obtained by fast Fourier transformation (FFT) of the pixel-averaged signal after performing Hann smoothing (with Hann as a window function to reduce spectral leakage) and normalization to the unit energy. The natural frequencies ($f$) and damping ratios ($\xi$) were estimated by fitting autocorrelation functions of waveforms using the following equation[77]:

$$R(\tau) = \frac{\pi G f_n e^{-2\pi f_n \xi|\tau|}}{4\xi k_s^2}\left(\cos(2\pi f_d \tau) + \frac{\xi}{\sqrt{1-\xi^2}}\sin(2\pi f_d |\tau|)\right) \qquad (4)$$

where $R$ is the autocorrelation function, $f_n$ is the undamped natural frequency, $f_d$ is the damped natural frequency, $\xi$ is the damping ratio, and $G$ is the input PSD amplitude.

## Methodological considerations

We employed k-means clustering to assist in seed point selection, but its efficacy is limited because it uses the Euclidean distance to evaluate functional connections and may not adequately address situations where FC patterns are comparable but exhibit varying strengths. Future research should focus on developing clustering algorithms more suitable for functional connectivity analysis. The areas of significant connectivity were calculated based on the $t$-test results using a $p$-value threshold of $p < 0.05$. However, the $t$-test is dependent on sample size and may exhibit variability, potentially introducing

inaccuracies in the areas of significant connectivity. The imaging method has certain limitations that need to be acknowledged. One limitation is that the employed chronic through-bone windows have inherent challenges in long-term usage, which restricts the ability to conduct longitudinal observations from infancy to adulthood on the same experimental animals. Our findings were confirmed by statistical analysis, which yielded consistent results. Additionally, the imaging method does not allow for precise identification of the specific cortical layer from which the observed fluorescence originates. Continued improvements in imaging techniques and animal preparation approaches will be necessary to overcome these limitations and further enhance our understanding of the intricate dynamics of the brain.

## Statistics and reproducibility

The animals are random selected. No data were excluded from the analyses. No statistical methods were used to pre-determine sample sizes. We imaged 6 VGLUT2 mice, 6 PV mice, 6 SOM mice, and 5 VIP mice at P14. At P28, we imaged 4 VGLUT2 mice, 7 PV mice, 6 SOM mice, and 7 VIP mice. At P56, we imaged 10 VGLUT2 mice, 13 PV mice, 10 SOM mice, and 10 VIP mice. We imaged 7 VGLUT2 mice at anesthesia state. Our sample sizes are similar to those reported in previous publications. Ages and anesthesia states cannot be concealed from the experimenters due to the evident differences in mouse size and condition.

## Reporting summary

Further information on research design is available in the Nature Portfolio Reporting Summary linked to this article.

## Data availability

The raw fluorescent images used in this study are available for download as NumPy files (.npy) without restrictions from: http://eai.brainsmatics.org/datasharing/shi2402. Source data are provided with this paper.

## Code availability

The code can be obtained with demo data at https://github.com/shih-liang/gssidnaim. All of our code used for this project is written in Python v3.10, making extensive use of Python packages, including NumPy v1.26.2, SciPy v1.9.1, statsmodels v0.13.5, matplotlib v3.8.2, and seaborn v0.12.2.

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

## Acknowledgements

This work was supported by the National Natural Science Foundation of China (NSFC) (61890951, 61890950, 62275095, and 82261138559), Fundamental Research Funds for the Central Universities, HUST

(2019kfyXMBZ009), Hainan University Research Start-up Fund (KYQD(ZR)20072 and KYQD(ZR)22074), the CAMS Innovation Fund for Medical Sciences (2019-I2M-5-014) and the Innovation Fund of WNLO. We thank the Optical Bioimaging Core Facility of WNLO-HUST for the support in data acquisition.

## Author contributions

L.S. led the analyses, helped with data registration and preprocessing, and co-wrote the manuscript; X.F. performed the surgeries, helped with data acquisition, registration and preprocessing, and co-wrote the manuscript; S.G. is responsible for the surgeries and the data acquisition; T.W. performed WGCNA; J.Z., J.L. and P.L. supervised the study and provided input on the manuscript. All authors contributed to reviewing and editing of the manuscript. L.S. and X.F. contributed equally.

## Competing interests

The authors declare no competing interests.
