## [Peer Review File · Nature Communications]

REVIEWER COMMENTS

Reviewer #1 (Remarks to the Author):

Shi et al. used wide-field calcium imaging to investigate the spatiotemporal structure of spontaneous activity in the mouse dorsal cortex under various conditions, including different developmental stages (P14, P28 and P56), different cell types (excitatory, PV, SOM and VIP-positive inhibitory neurons) and anesthesia/awake states. The authors further showed some correspondences between spatial patterns of cortex-wide activity and those of gene expression.

While the central ideas and approaches in this study largely build upon earlier fMRI studies in humans (e.g., Bolt et al. 2022), the authors do not sufficiently elaborate on their analytical methods in the main text. The lack of detail, together with the absence of rigorous statistical analysis, raises concerns about the validity of their findings. A potentially novel aspect of this study lies in its exploration of cortical activity across different cell types and developmental stages. However, the findings seem to be relatively marginal, particularly in light of existing literature that has documented large-scale cortical activity in mice under various contexts and cell types (Wekselblatt et al. 1996; Allen et al. 2017; Makino et al. 2017; Musall et al. 2019; Barson et al. 2020; Gallero-Salas et al. 2020; Lake et al. 2020; Ren et al. 2022). Moreover, the relationship between neural activity and gene expression has also been reported previously (Richiardi et al. 2015; Wang et al. 2015).

In summary, although the paper offers a comprehensive overview of large-scale cortical activity at resting states in mice, its contribution to the field appears to be incremental in my opinion (though other reviewers may disagree) and its lack of rigorous statistical validations further impede a full evaluation of the results.

Major:

1. I believe that the authors analyzed the imaging data based on the methods described in Bolt et al. 2022, which studied resting-state BOLD signals in humans. However, the absence of detailed descriptions regarding their analysis limits the manuscript's accessibility and interpretability within the neuroscience community.

2. In Figure 2f, the authors investigated the relationship between φ_4 and asymmetric resting-state functional connectivity. I assume that wave numbers were assigned according to the order of explained variance. Hence, the asymmetrical activity patterns could, in theory, appear at various φ depending on the cell type and developmental stage. This is evident in Figure S3, where, for example, the φ_4 pattern

for VGLUT2 at P14 does not exhibit noticeable asymmetry. Thus, the decision to uniformly choose φ_4 for all conditions seems unjustifiable.

3. In Figure 5a-c, the authors identified φ_2 , φ_4 and φ_5 but these wave numbers are not presented in the order of explained variance. For example, the explained variance for φ_2 in PV at P14 appears negligible. How were these wave numbers determined? In addition, the arrangement of φ_2 at P56 seems wrong, as indicated by the yellow box being placed second from top. The spatial activity patterns within each column also show discrepancies in several instances.

4. The study's focus on developmental processes across different cell types offers a potentially novel finding that has not been covered in human studies. However, the manuscript fails to elucidate how these unique developmental trajectories influence the activity patterns observed in other cell types.

5. In Figure 7a, it is indicated that the activity patterns featured in Figure 7d, termed "DMN-associate RSWs" are prevalent under anesthesia. However, these patterns are not placed in the top rows. Why? Could the observed decrease in RSW proportion be primarily explained by the increase in "DMN-associate RSWs"? What do these colors mean and why are there five of them?

6. In most cases, the manuscript lacks statistical descriptions (e.g. error bars, p-values), making it difficult to evaluate their results.

Minor:

1. What does the scale bar in Figure 1b represent?

2. There are several typos (line 113 "fifth", line 132 "third", in 335 "tempo", line 436 "voxels").

3. I could not find legends for the supplementary figures.

4. I suggest that the authors reference relevant mouse studies.

5. In line 283, what is φ_0 ?

6. There is no legend for Figure 8c.

7. The methods section requires further elaboration.

Reviewer #2 (Remarks to the Author):

Exploring the neurophysiological relevance and origin of resting state activity, or better termed task-free spontaneous activity represents an unmet need which hinders the interpretation of rs functional connectivity data particularly in human fMRI studies. Shi and colleagues conduct a technically and analytically challenging study using wide-field functional calcium imaging of mouse cortex. What is more, the authors conduct a cell-specific analysis by restricting expression of GCaMP to various cell types, vigilance states and even developmental stages. They put forward an interesting analysis concept, differentiating between standing and traveling waves of activity. While the analysis and the technical execution is state of the art and sound, I see major limitations:

1. My main criticism is the almost complete lack of conceptual understanding of spontaneous activity: the term “resting state” originated from the discovery of infra-slow signal fluctuations in fMRI studies. And almost all literature cited by the authors stem from fMRI RS studies. While there is mounting evidence, that indeed RS fMRI signal do have a neurophysiological correlate, here, the terminology is simply false: Here, the authors study (almost) direct neuronal activity, with a temporal resolution of 30 Hz. This is not a mere terminology issue, but goes much further: Maybe most striking, in line 300 “Our finding reveal that even during movement,..., ..components resemble those of the resting state”- Of course they do! The authors measure ongoing spontaneous activity, the brain is continuously active, also during movement. I would refer the authors to the vast amount of literature on spontaneous activity, maybe starting by the seminal work of Kenneth Harris, and Carl Petersen. And most certainly, movement impacts spontaneous activity (please see the seminal work of Stryker and colleagues), but it is still spontaneous neuronal activity.
2. Along these lines, the authors state that they are studying “anesthesia”, and term the state “burst suppression”. Again, I would refer the authors to the pioneer of studying spontaneous activity, the late Steriade, and continued by McCormick and Mavi Sanchez Vives, as these are slow oscillations, as clearly seen in their data (Fig 5 d). In slow wave state, or deep sleep, pancortical synchronized activity dominates, which also includes travelling calcium waves
3. I could go on and on, there is a complete lack of understanding of the neurophysiological research on spontaneous task free activity, as e.g. evident in line 68 “..implying a close connection between DMN and burst suppression”. What does that mean?
4. Also on the developmental angle, certainly it is worthwhile studying developmental aspects in spontaneous activity, but the time points used are meaningless from a developmental viewpoint, as all

time points are after eye opening (occurring around P 12), and after the GABA switch, so if they aim for studying these effects, they would have to study earlier time points.

5. Lastly, also the cell-type specific analysis lack any understanding of the basic connectivity schemes of the cortex.

Reviewer #3 (Remarks to the Author):

In NCOMMS-23-37172, Shi et al investigate the neural underpinnings of so-called “resting-state waves” using calcium imaging with cell-type specificity (multiple cell types) in awake mice. The authors explored the spatiotemporal patterns emerging from such recordings under multiple conditions, including different anesthesia states, different developmental stages, nonrusting conditions, and also correlated extracted spatial patterns with gene expression. The authors verify that these waves underpin resting-state connectivity, and then also show that four different types of neurons (both excitatory and inhibitory) show similar global patterns. Interestingly, the developmental study showed varying timelines in the evolution of the waves for each cell type, which is a deep and interesting finding in my opinion. In non-resting conditions, waves were still decomposable and different to some extent compared with resting state waves.

Overall, this work represents a significant advance in our understanding of global brain activity patterns, which these days is becoming an increasingly “hot topic” (e.g. Pang et al, Nature 2023). In my view, the main advance here is the introduction of cell-type specificity in these measurements, which reveal the neural origins of these signals, as well as the developmental aspects of the study which I find both highly novel and intriguing in terms of the actual results. The study would certainly be of value to a broad audience and would appeal to Nat Comm readership.

I do have a few comments that I think would strengthen the work substantially:

Major Comments

1. Refinement of the novel aspects of the work

As mentioned above, there are many novel aspects to this work. However, I find some of the claims exaggerated. For instance, the subheading “Global structures of resting-state activity exist in mice” is a large overstatement and not per-se novel. These structures have been observed before, either indirectly by resting-state fMRI in mice (CAPs and QPPs for example, are “snapshots” of this kind of activity), and even in optical imaging experiments (e.g. Ma et al, Resting-state hemodynamics are spatiotemporally coupled to synchronized and symmetric neural activity in excitatory neurons, PNAS 2016 and papers thereafter). In addition, a recent study (Cabral J et al, Intrinsic macroscale oscillatory modes driving long range functional connectivity in female rat brains detected by ultrafast fMRI, Nature Communications 2023) revealed these waves in rats.

I think the authors could forego “forcing” novelty on the existence of the waves, and they probably should discuss (and cite) a few more papers that observed similar CAPs / QPPs and discuss their findings with relation to the prior art. This will not detract from the other impressive novel features presented in the work.

2. Temporal aspects in the wave analyses

The authors did a great job in visualizing and comparing the spatial parts of the waves. However, it seems to me that the temporal aspects have been quite overlooked, which I believe is a large missed opportunity which would greatly enhance the study's relevance and impact.

The authors should deeply analyze the time series; they probably ought to observe damped oscillatory nature of these signals (c.f Cabral et al Nat Comm 2023), and so they could quantify frequency of activation along the different conditions (cell types, developmental stage), and further attempt to extract how frequently each of these patterns exist in each condition. This will reveal how often the brain “visits” these modes (so to speak).

I strongly recommend to look into the temporal aspects more deeply here.

3. Comparison of not-necessarily overlapping wave patterns

The authors compare waves across different conditions. But it seems like waves have varying spatiotemporal characteristics (e.g. ϕ_1 in condition A could be different from ϕ_1 in condition B; or, there could be a different number of underlying waves contributing to the total signal between conditions). How was this taken into account?

4. Unclear figures and missing statistical comparisons

4a. There are several figures that are difficult to understand. Most prominently, perhaps, Fig 7c: what are conditions 1-12? 12 waves? This should really be spelled out.

4b. In addition, Fig 7a, the dashed lines are confusing – what do they represent?

4c. Fig 2c should be explained in the text.

4d. There are no error bars on Fig 2b, 2d and 2f, and statistical significance is not indicated. Please quantify.

4e. How were multiple comparisons accounted for in this work?

4f. Add heatmap to Fig 4b

4g. In Fig 1, is phi3 deliberately omitted?

4h. Figure 4c and 4d: why are there two of every line color? This figure needs to be explained much better in the text and in the caption.

5. Sensitivity of different cell-type markers

Do the authors expect any differences in sensitivity with each cell-type calcium reporter? Could differing expression patterns/efficiency of each marker have played a role in some of the findings in the work?

6. Connection with fMRI

The impact of the work here could be dramatically enhanced if the authors analyzed their data convolved with a hemodynamic response function, and provided insight into rs-fMRI studies from the four cell types they recorded. Something akin to what Ma et al PNAS 2016 did would really boost the impact of this work, especially if discussed in connection with CAPs, QPPs and Intrinsic Oscillatory Modes.

Minor Comments

6. The paper could benefit from heavy editing. The Introduction section is (paradoxically) both not concise and incomplete.
7. In addition, the text from line 61 to the end of the introduction, is actually a mini results section. It should be heavily edited and perhaps even completely removed.
8. Line 107: it would be probably worthwhile to adhere to already accepted nomenclature such as “QPP” or “intrinsic oscillatory mode”.
9. Figure 2f, the yellow legend for VGLUT2 slipped downward and is overlapping with the green SOM legend. Please fix.
10. Line 148: “... having exhibiting average” – unfinished sentence.
11. Fig.4 – why not add P14->56?
12. RSW is a somewhat paradoxical name as you find them under activity too. Perhaps go with Intrinsic Macroscale Oscillatory Modes like Cabral et al 2023?
13. Lines 319-322: this is arguably one of the most important findings of this work. Suggest expanding on this.
14. line 335: “tempo spatial” is a misnomer. Either temporospatial or spatiotemporal. Also, at this place, it would be good to connect to Cabral et al Nat Comm 2023, with their ultrafast acquisitions.
15. Lines 342-346: too speculative. Suggest removing altogether.

We thank the reviewers for their thoughtful, constructive and detailed feedback on our manuscript. In response to the specific concerns regarding the interpretation of our findings, we have conducted an extensive re-evaluation of our data and revised our interpretations to more accurately reflect the results particularly in Discussion. Additionally, in line with the suggestions of reviewers 1 and 2, we have provided more relevantly context in Introduction and added rigorous statistical validations in figures. And we conducted comprehensive temporal analysis advised by reviewer 3. These additions not only address the concerns raised but also enrich the overall depth and breadth of our research. We have added new data in **Figures 1-3, 5-8** and Supplementary **Figures 1-5** and edited the main text in responds to reviewer comments (as detailed below); these changes are **highlighted in red** in the revised manuscript and the response. In our point-by-point response to reviewers below, the reviewers' remarks are presented in **blue** and our responses are shown in black text.

Reviewer #1 (Remarks to the Author):

Shi et al. used wide-field calcium imaging to investigate the spatiotemporal structure of spontaneous activity in the mouse dorsal cortex under various conditions, including different developmental stages (P14, P28 and P56), different cell types (excitatory, PV, SOM and VIP-positive inhibitory neurons) and anesthesia/awake states. The authors further showed some correspondences between spatial patterns of cortex-wide activity and those of gene expression.

While the central ideas and approaches in this study largely build upon earlier fMRI studies in humans (e.g., Bolt et al. 2022), the authors do not sufficiently elaborate on their analytical methods in the main text. The lack of detail, together with the absence of rigorous statistical analysis, raises concerns about the validity of their findings. A potentially novel aspect of this study lies in its exploration of cortical activity across different cell types and developmental stages. However, the findings seem to be relatively marginal, particularly in light of existing literature that has documented large-scale cortical activity in mice under various contexts and cell types (Wekselblatt et al. 1996; Allen et al. 2017; Makino et al. 2017; Musall et al. 2019; Barson et al. 2020; Gallero-Salas et al. 2020; Lake et al. 2020; Ren et al. 2022). Moreover, the relationship between neural activity and gene expression has also been reported previously (Richiardi et al. 2015; Wang et al. 2015).

In summary, although the paper offers a comprehensive overview of large-scale cortical activity at resting states in mice, its contribution to the field appears to be incremental in my opinion (though other reviewers may disagree) and its lack of rigorous statistical validations further impede a full evaluation of the results.

Reply:

Thank you very much for the time you spent on reading and for the detailed critiques. In response to the concerns on method, we have extensively elaborated our analytical method of CPCA in the legend of Fig 1

and the Methods section of revised manuscript, including the detailed description of normalization, Hilbert transformation, PCA and group-level analysis (as demonstrated in the response of major points). We have also performed statistical analyses and added statistical descriptions such as error bars and p-values where feasible. And we have done a better job at highlighting our novelty by providing more context and comparisons in the Introduction and Discussion sections. As the reviewer can find, the Introduction and Discussion sections have almost been re-written to clarify the novelty. Compared to large-scale cortical activity documented in mice under various contexts and cell types, we portray a highly multidimensional continuum of synchronizations that includes time-lag synchronization, which may initially be interpreted as a desynchronized state, and differences in synchronization between cells. New in text:

The spontaneous neural activity of the brain demonstrates self-organized intrinsic dynamics¹⁻⁷, which is closely associated with stimulation, cognition, and behavior, ranging from highly synchronized to desynchronized⁸. Functional connectivity (FC) has been commonly employed to depict the spatial organization of the brain synchronization and is recognized for its alterations during development and across pathological conditions⁹⁻¹⁴. However, mounting evidence, gathered at various states including wakefulness using various techniques ranging from cellular to whole-brain scales, has revealed the widespread presence of nonstationary synchronization¹⁵⁻²⁴. These phenomena may encompass terms like spatiotemporal patterns, standing/traveling waves, zero-lag/time-lag synchronies and exhibit regional specificity at large scales. Their regional specificity may not align with brain regions or the structural connectome^{25,26}. Moreover, FC, as a factor of stationary synchronization, is influenced by nonstationary synchronization, suggesting that some phenomena previously interpreted as desynchronization might in fact be nonstationary synchronization.

Complex global synchronization structures in various cell types, ages, and physiological states. These structures, present in the spontaneous activity of the brain, show a certain level of conservation across different cell types, ages, and even species, such as humans and mice. Although state-dependent FC has been identified in various contexts, the synchronization of global neural activities extends beyond simply something between synchronized or desynchronized. It encompasses a more intricate picture as a highly multidimensional continuum of synchronizations including time-lag (may be interpreted as desynchronized state early) and differences between cells.

Our study also addresses an unmet need which hinders the interpretation of rs functional connectivity data particularly in human fMRI studies:

Global signal regression (GSR) is a controversial method commonly employed in fMRI studies⁴⁴. Previous studies have suggested that global signal regression (GSR) can introduce a more pronounced negative correlation in FC⁴⁵. Compared to fMRI and ultrafast fMRI⁴⁰, our method offers finer spatiotemporal resolution and enables almost direct imaging of intracellular signals, unaffected by the HRF, potentially capturing more characteristics of neural activities. Leveraging fluorescent signals, we investigated the impact of GSR on FC. Our analysis revealed that while GSR eliminates zero-lag synchronization, it preserves FC structures, which reflect time-lag synchronization. Our analysis revealed negative correlations in spontaneous neural activity, arising from the propagation of specific wave patterns. These

interferences can lead to near-zero correlation coefficients between activity in different brain regions, some of which may be interpreted as desynchronization previously. These findings address a gap in understanding of resting-state FC (rsFC) data, particularly in human fMRI studies.

Major:

1. I believe that the authors analyzed the imaging data based on the methods described in Bolt et al. 2022, which studied resting-state BOLD signals in humans. However, the absence of detailed descriptions regarding their analysis limits the manuscript's accessibility and interpretability within the neuroscience community.

Reply:

We agree with the reviewer that original description limits the manuscript's accessibility and interpretability. We have added more details in the Method section:

The CPCA method involves applying a Hilbert transform to the original signal, followed by principal component analysis (PCA). Our fluorescent signal at each pixel (p) is a time (t) series $S_p(t)$. For each S_p , it is first z-scored ($N_p = \frac{S_p - \text{Mean}(S_p)}{\sqrt{\text{Var}(S_p)}}$), then subjected to a Hilbert transform, $C_p = \text{Hilbert}(N_p)$. The outcome of the Hilbert transform is a complex number, forming a matrix:

$$M = \begin{pmatrix} C_{p_0}(t_0) & \cdots & C_{p_N}(t_0) \\ \vdots & \ddots & \vdots \\ C_{p_0}(t_M) & \cdots & C_{p_N}(t_M) \end{pmatrix} \quad (1)$$

Principal component decomposition of M yields scores ($t_k(t)$ for the k -th component), eigenvalues (d_k for the k -th component) and weights ($w_k(p)$ for the k -th component). w_k represents the spatial distribution of standing/traveling waves (complex numbers, including amplitude and phase distribution). The real part of t_k represents the waveforms of standing/traveling waves, while d_k^2 represents variances of each wave. Combining scores and weights, or projecting the original signal with weights, yields the spatiotemporal pattern of each standing/traveling wave (Supplementary Video).

For group level analysis, we normalize the signals obtained from each experiment $N1_p, N2_p, \dots$ and then concatenate them over time to form a longer signal ($NG_p = [N1_p(t_0), \dots, N1_p(t_n), N2_p(t_0), \dots, N2_p(t_n), \dots]$), then repeat the same steps as in a single experiment. The effectiveness of CPCA depends on the length of the signal; generally, the longer the signal, the higher the signal-to-noise ratio in the results.

2. In Figure 2f, the authors investigated the relationship between ϕ_4 and asymmetric resting-state functional connectivity. I assume that wave numbers were assigned according to the order of explained variance. Hence, the asymmetrical activity patterns could, in theory, appear at various ϕ depending on the cell type and developmental stage. This is evident in Figure S3, where, for example, the ϕ_4 pattern

for VGLUT2 at P14 does not exhibit noticeable asymmetry. Thus, the decision to uniformly choose ϕ_4 for all conditions seems unjustifiable.

3. In Figure 5a-c, the authors identified ϕ_2 , ϕ_4 and ϕ_5 but these wave numbers are not presented in the order of explained variance. For example, the explained variance for ϕ_2 in PV at P14 appears negligible. How were these wave numbers determined? In addition, the arrangement of ϕ_2 at P56 seems wrong, as indicated by the yellow box being placed second from top. The spatial activity patterns within each column also show discrepancies in several instances.

Reply to 2,3:

Thank you for pointing this out. Using numbers as wave names make it harder to understand. We now give names to each wave names: Φ_G , Φ_{SM1} , Φ_{SM2} , Φ_T and $\Phi_{DMN-like}$ and detailed described how and which these waves in each cell type and conditions selected in supplementary Fig 3. Here's the brief:

We compared the correlation matrices of all waves from various cell types in different states with Φ_0 , Φ_1 , Φ_2 , Φ_4 and Φ_5 from VGLUT2 at P56. We identified those with the highest similarity as belonging to the same category. The figure displays the correlation matrices of the waves with the highest similarities. The text below each matrix indicates the wave and its similarity to the corresponding wave in VGLUT2, measured by mean square error (MSE), with lower values indicating better similarity.

There is no direct method to compare different waves from CPCA results. To identify corresponding waves, we examine the connectivity characteristics of the waves (C_{Φ_i}). We start with the spatiotemporal pattern of each wave and compute its connectivity matrix, i.e., the correlation coefficients between pixels. We then compare these connectivity matrices based on mean square error, considering the waves with the lowest mean square error as the same type. This mean square error is shown in Supplementary Fig 3.

Supplementary Fig 3. **Correlation matrices of reconstructed time courses from standing/traveling waves.** (a-e) Correlation matrices of reconstructed time courses from Φ_G , Φ_{SM1} , Φ_{SM2} , Φ_T and $\Phi_{DMN-like}$. We compared the correlation matrices of all waves from various cell types in different states with Φ_0 , Φ_1 , Φ_2 , Φ_4 and Φ_5 from VGLUT2 at P56. We identified those with the highest similarity as belonging to the same category. The figure displays the correlation matrices of the waves with the highest similarities. The text below each matrix indicates the wave and its similarity to the corresponding wave in VGLUT2, measured by mean square error (MSE), with lower values indicating better similarity.

4. The study's focus on developmental processes across different cell types offers a potentially novel finding that has not been covered in human studies. However, the manuscript fails to elucidate how these unique developmental trajectories influence the activity patterns observed in other cell types.

Reply:

We have extended the Discussion section for this:

Different developmental trajectories in FC and traveling waves are observed for GABAergic interneurons and glutamatergic neurons. FC of SOM and glutamatergic are established at an early stage, while FC of PV establishes connections throughout mouse development, and rapid expression of PV and establishment of synapses being among the most critical changes in the GABAergic system during this period⁴⁸. FC of VIP establishes after adolescence. Despite VIP expression being abundant during embryonic and early life stages⁴⁹, it remains unclear why its FC is mainly established between P28 and P56. VGLUT2 and SOM neurons establish relatively complete synchronizing structures at early stages. While VGLUT2 neurons exhibit nearly perfect time-lag synchronization even at early stages, GABAergic neurons except SOM lack this type of synchronization. Studies indicate that neural desynchronization may stem from shared input/output among excitatory and inhibitory neurons^{1,2}. The traveling wave robustness in VGLUT2 at early age presents a challenge in the mechanisms for time-lag synchronizations due to the immaturity of GABAergic neurons.

Interaction between different types of neurons. We observed opposite developmental trajectories between PV and VGLUT2 in traveling waves. It is known that PV expression and rapid synapse formation occur during this period⁴⁸ and PV inhibit glutamatergic neurons directly⁴⁶. The changes in VGLUT2 traveling waves from P14 to P56 may be associated with the establishment of PV synchronization. PV cells inhibit the cell bodies and proximal dendrites of pyramidal cells, while SOM cells target the distal dendrites⁵⁸⁻⁶⁰. Horizontal propagation in layer 5 pyramidal cells during the up state mainly occurs among their cell bodies, and thus, inhibition by PV cells at the cell bodies is likely the primary inhibitory system, with SOM cell inhibition of distal dendrites acting as a secondary system⁶¹. However, the impact of cell type on time-lag synchronization remains unclear. Notably, the timing of mature synchronization structures differs across cell types. After P28, VIP synchronization matures rapidly, while VGLUT2 synchronization exhibits a seemingly opposite pattern to its pre-P28 behavior. Before P28, traveling waves in VGLUT2 decrease coincides with the rise of that in PV. However, after P28, the traveling waves of VGLUT2 align with those of both PV and VIP, coinciding with modifications in the presynaptic terminal during this period⁶², resulting in a more complex picture. SOM neurons show early developed, stable synchronizing structures. In neural systems, excitatory activity typically triggers balanced inhibition for stability⁶³. VGLUT2 neurons rapidly mature during the early stages of eye opening, and SOM neurons form synaptic connections with excitatory neurons before P14, leading to early and relatively complete synchronizations similar to those seen in VGLUT2 neurons.

5. In Figure 7a, it is indicated that the activity patterns featured in Figure 7d, termed "DMN-associate

RSWs” are prevalent under anesthesia. However, these patterns are not placed in the top rows. Why? Could the observed decrease in RSW proportion be primarily explained by the increase in “DMN-associate RSWs”? What do these colors mean and why are there five of them?

Reply:

We have added Fig 8a to address this issue, as our previous manuscript lacked clarity in its explanation. The default mode network (DMN)-associated resting-state waves (RSWs) shown in Fig 7a, now referred to as traveling waves $\Phi_{DMN-like}$, are extracted through covariance. The activity patterns featured in Fig 7d (Fig 8b now) are co-activation patterns representing raw neural activity.

Under anesthesia or in slow-wave states, calcium fluorescence exhibits slow waves. The raw neural activity in Figure 8b reflects these slow waves under anesthesia. The slow wave was also extracted by CPCA, denoted as Φ_0 , or Φ_G , shown in Figure 8a. It is distinct from $\Phi_{DMN-like}$. We omitted Φ_G in the previous version, leading to confusion. We have now clarified this distinction and addressed this through Fig 8.

Fig 8 a, b. **Standing waves and frequency characteristics in VGLUT2 neurons during anesthesia.** (a) Spatiotemporal patterns of waves Φ_G during anesthesia, at BSR level of 90%. (b) Prominent co-activation patterns in anesthetized state (90% BSR).

Co-activation patterns (CAPs) during anesthesia are predominantly characterized by alternating activations between central and peripheral cortical regions, as illustrated in Fig. 8b. Correspondingly, the standing wave Φ_G under anesthesia (Fig. 8a) shows not just overall changes but also detailed variations. Notably, there is a wave that propagates from the retrosplenial area (RSP) to the lateral cortex or vice versa under anesthesia, varying with different anesthetics, as seen in supplementary videos. The spatial pattern of Φ_G closely resembles the main pattern of CAPs. Moreover, the global signal also contains traveling components, which are not addressed by global signal regression (GSR), potentially highlighting a limitation of GSR.

A complex picture of brain synchronization under anesthesia. Anesthesia can induce the brain to enter a more synchronized state, a slow-wave state similar to sleep. In this state, pancortical synchronized activity is dominant, as evidenced by our results as Φ_0 . However, we observed the presence of traveling waves other than these slow oscillations, suggesting a more complex synchronization state, which consist with earlier studies. Furthermore, although all slow oscillations (all Φ_0 under anesthesia) exhibit almost zero-lag synchronized activity, slow waves induced by different anesthetics display distinct characteristics in their spatial movement at small scale. As anesthetics typically affect the GABA system, this might be a comprehensive outcome, related to the mechanisms of synchronization including the interactions between GABAergic and glutamatergic neurons. This phenomenon warrants further investigation.

6. In most cases, the manuscript lacks statistical descriptions (e.g. error bars, p-values), making it difficult to evaluate their results.

Reply:

Thank you for pointing out. We reviewed the figures and added statistical descriptions when feasible. For example:

Fig. 1: Large-scale neural activity is organized by standing and traveling waves (VGLUT2, P56). (a) Fluorescent signals $S(t)$ of neural activity can be decomposed into a linear superposition of standing and travelling waves $\Phi_i(t)$ distributed throughout the cortex, which are extracted from spontaneous neural activity in VGLUT2 neurons at P56. The weight w_i is eigenvalue of Φ_i , $w_i\varphi_i$ gives the principal component “scores”, and $(w_i\rho_i e^{\theta_i}) / \sqrt{n_{time} - 1}$ gives the principal component “loadings”. (b) Spatiotemporal patterns (over one cycle) of waves $\Phi_0, \Phi_1, \Phi_4, \Phi_5$, showcasing only significant waves identified in the subsequent text. (c) Spatial distributions of waves $\Phi_0, \Phi_1, \Phi_4, \Phi_5$. ρ : Intensity distribution. θ : Phase distribution (time lag) in rad. (d) The power spectral density (PSD) of the fluorescence signal (pixel averaged, normalized to unit energy). (e) Spatial maps of spectral power in 6 non-overlapping frequency bands, with pixels normalized to unit energy. (f) Waveforms $\varphi_i(t)$ and autocorrelation functions of waves $\Phi_0, \Phi_1, \Phi_4, \Phi_5$. (g) Unit-energy PSDs of $\varphi_0, \varphi_1, \varphi_4, \varphi_5$, with inter-experiments variability (shaded). Nature frequencies (f) and damping ratios (ξ) of 10 strongest waves. (h) The proportion of the strongest 10 waves, measured by the variance ratio of each wave relative to the original signal. (i) The spatial distribution uniformity of the strongest 10 waves, measured by the circle variance of θ_i .

Fig. 2: FC as a linear superposition of zero-lag and time-lag synchronization structures. (a) FC matrices were calculated for original fluorescent signals (FC), GSR-regressed signals (FC_{GSR}), and reconstructed time courses from Φ_0 (C_{Φ_0}), Φ_1 (C_{Φ_1}), Φ_4 (C_{Φ_4}) and Φ_5 (C_{Φ_5}). While the values differ considerably, FC matrix and FC_{GSR} matrix display a remarkable correlation ($r = 0.93 \pm 0.03$), suggesting a shared underlying trend in their relative positioning. (b) Similarities between FC and FC_{GSR} maps with C_{Φ_i} , including individual and their linear superpositions ($\sum_{t=0}^i w_t C_{\Phi_t}$), measured by Pearson correlation coefficient. (c) Spatial distributions of pixel-averaged FC (D_{FC}) and $|FC_{GSR}|$ ($D_{FC_{GSR}}$) with their hemispheric differences (Δ_{FC} , $\Delta_{FC_{GSR}}$). The hemispheric differences are calculated by pixel average of $FC_{Homolateral} - FC_{Contralateral}$. (d) The similarities between spatial distribution of FC (D_{FC} , $D_{FC_{GSR}}$) and the intensity distributions of standing/traveling waves (strongest first 10 waves, ρ_i as in Fig 1c), measured by Pearson correlation coefficient, only positive correlations shown. D_{FC} and ρ_0 , $D_{FC_{GSR}}$ and ρ_1 , Δ_{FC} , $\Delta_{FC_{GSR}}$ and ρ_4 exhibited significantly higher pairwise similarities than all other combinations (Fisher's z-transformed t-tests, $p < 10^{-6}$, $p < 6.88 \times 10^{-6}$, $p < 8.87 \times 10^{-3}$ after FDR correction with threshold of 0.05).

Minor:

1. What does the scale bar in Figure 1b represent?

Reply:

It's the frequency that CAP changes. We have now removed Figure 1b now for highlighting our novel result.

2. There are several typos (line 113 "fifth", line 132 "third", in 335 "tempo", line 436 "voxels").

Reply:

Revised. Thank you.

3. I could not find legends for the supplementary figures.

Reply:

Thanks for pointing out. We have revised all supplementary figures and corrected all legends.

4. I suggest that the authors reference relevant mouse studies.

Reply:

We have provided a more comprehensive context in the introduction and discussion, and referenced more mouse studies.

5. In line 283, what is ϕ_0 ?

Reply:

Thanks for pointing out. We fixed this typo as Φ_1 .

6. There is no legend for Figure 8c.

Reply:

Thanks for pointing out. The missing legend is now at place:

Fig 9c. Biological processes associated with genes that are correlated with traveling waves.

7. The methods section requires further elaboration.

Reply:

We agree and have rewritten the methods section with more details

Reviewer #2 (Remarks to the Author):

Exploring the neurophysiological relevance and origin of resting state activity, or better termed task-free spontaneous activity represents an unmet need which hinders the interpretation of rs functional connectivity data particularly in human fMRI studies. Shi and colleagues conduct a technically and analytically challenging study using wide-field functional calcium imaging of mouse cortex. What is more, the authors conduct a cell-specific analysis by restricting expression of GCaMP to various cell types, vigilance states and even developmental stages. They put forward an interesting analysis concept, differentiating between standing and traveling waves of activity. While the analysis and the technical execution is state of the art and sound, I see major limitations:

Reply:

Thank you very much for the positive assessment of our work and for constructive and detailed feedback. We have made great efforts to improve our manuscript based on your comments. We have read the relevant literature on spontaneous activity, addressed the terminology issues, improved the Introduction and Results sections, and discussed the cell-type specific analysis in more detail to make the manuscript more rigorous and accurate.

1. My main criticism is the almost complete lack of conceptual understanding of spontaneous activity: the term “resting state” originated from the discovery of infra-slow signal fluctuations in fMRI studies. And almost all literature cited by the authors stem from fMRI RS studies. While there is mounting evidence, that indeed RS fMRI signal do have a neurophysiological correlate, here, the terminology is simply false: Here, the authors study (almost) direct neuronal activity, with a temporal resolution of 30 Hz. This is not a mere terminology issue, but goes much further: Maybe most striking, in line 300 “Our finding reveal that even during movement,..., ..components resemble those of the resting state”- Of course they do! The authors measure ongoing spontaneous activity, the brain is continuously active, also during movement. I would refer the authors to the vast amount of literature on spontaneous activity, maybe starting by the seminal work of Kenneth Harris, and Carl Petersen. And most certainly, movement impacts spontaneous activity (please see the seminal work of Stryker and colleagues), but it is still spontaneous neuronal activity.

Reply:

Thank you for your insightful critiques. We have reviewed relevant literature, including the works you mentioned, and have made significant improvements to our manuscript accordingly. We acknowledge and agree that the distinction between spontaneous neural activity and resting-state fMRI signals is not merely a terminological issue. In our revised manuscript, we have taken greater care to differentiate between brain states and signals and deleted a lot of fuzzy or wrong expressions.

Our analysis now includes a more detailed examination of the characteristics of both waves and functional connectivity (FC) in describing synchronization (Fig 2, traveling waves stands for time-lag synchronization and FC is superposition of zero-lag and time-lag synchronizations). We acknowledge and agree that the brain is continuously active, including during movement, and movement impacts spontaneous activity. We are interested in understanding the large-scale changes that occur as a result and analyzed the difference in the revised manuscript (Fig 6). We found during movement, the uncorrelatedness between some waves has disappear (Fig 6c).

Fig. 2: FC as a linear superposition of zero-lag and time-lag synchronization structures. (a) FC matrices were calculated for original fluorescent signals (FC), GSR-regressed signals (FC_{GSR}), and reconstructed time courses from Φ_0 (C_{Φ_0}), Φ_1 (C_{Φ_1}), Φ_4 (C_{Φ_4}) and Φ_5 (C_{Φ_5}). While the values differ considerably, FC matrix and FC_{GSR} matrix display a remarkable correlation ($r = 0.93 \pm 0.03$), suggesting a shared underlying trend in their relative positioning. (b) Similarities between FC and FC_{GSR} maps with C_{Φ_i} , including individual and their linear superpositions ($\sum_{t=0}^i s_t C_{\Phi_t}$, s_t is the variance explained by the t -th wave), measured by Pearson correlation coefficient. (c) Spatial distributions of pixel-averaged FC (D_{FC}) and $|FC_{GSR}|$ ($D_{FC_{GSR}}$) with their hemispheric differences (Δ_{FC} , $\Delta_{FC_{GSR}}$). The hemispheric differences are calculated by pixel average of $FC_{Homolateral} - FC_{Contralateral}$. (d) The similarities between spatial distribution of FC (D_{FC} , $D_{FC_{GSR}}$) and the intensity distributions of standing/traveling waves (strongest first 10 waves, ρ_i as in Fig 1c), measured by Pearson correlation coefficient, only positive correlations shown. D_{FC} and ρ_0 , $D_{FC_{GSR}}$ and ρ_1 , Δ_{FC} , $\Delta_{FC_{GSR}}$ and ρ_4 exhibited significantly higher pairwise similarities than all other combinations (Fisher's z-transformed t-tests, $p < 10^{-6}$, $p < 6.88 \times 10^{-6}$, $p < 8.87 \times 10^{-3}$ after FDR correction with threshold of 0.05).

Fig. 6: Extracted waves by CPCA during movement. (a) Spatial distributions of the waves Φ_G , Φ_{SM} during movement (running on plate) in four types of neurons. (b) Changes in wave quality during movement, measured by PSNR between the spatial distribution and its median filtering. The quality of strongest 10 waves are presented. (c) Spatial distribution and correlation matrix of Φ_4 during movement.

During the movement, several components are similar to those observed at rest (Supplementary Fig. 1), but there are waves severely deformed. As shown in Fig 6c, the correlation matrix of Φ_4 during the movement appeared to be the combination of Φ_T and $\Phi_{DMN-like}$ at rest (Fig 2), with MSE of 0.756 and 0.831 respectively (Supplementary Fig. 3), which means the uncorrelatedness between these waves has disappear. The change of waves in the four types of neurons during movement is different, and deterioration of “quality” is a common phenomenon. PV and VGLUT2 are the most affected, SOM is the least affected, and VIP is in the middle (Fig 6b). This may be due to the differences in the roles of different cells during spontaneous movement.

2. Along these lines, the authors state that they are studying “anesthesia”, and term the state “burst suppression”. Again, I would refer the authors to the pioneer of studying spontaneous activity, the late Steriade, and continued by McCormick and Mavi Sanchez Vives, as these are slow oscillations, as clearly seen in their data (Fig 5 d). In slow wave state, or deep sleep, pancortical synchronized activity dominates, which also includes travelling calcium waves

Reply:

Thank you for your valuable feedback. In our study, we examined conditions under anesthesia at different Burst Suppression Ratio (BSR) levels. Although the concentration of anesthetics is a factor for the depth of anesthesia, considering individual differences, BSR is often used as an indicator for anesthesia monitoring in clinical practice. Hence, we followed clinical methodologies in our research. We have added this information to the main text and have been more meticulous in our choice of terminology, as shown below. The presence of a slow wave state and more synchronized activity indeed occurs in the anesthetized state, as demonstrated in our results. We observed that the directionality of slow waves differs between anesthetized and awake states.

We concur that under anesthesia or in slow-wave states, calcium fluorescence shows slow waves. The slow waves were also extracted by CPCA, denoted as Φ_0 or Φ_G , as shown in Figure 8a. Other waves like Φ_{SM} , Φ_{SM2} and $\Phi_{DMN-like}$ are distinct (zero correlation with) from Φ_0 . There are other waves except dominating pancortical synchronized activity. In our previous version, we inadvertently omitted Φ_G , which may have caused some confusion. We have now rectified this by clarifying the distinction and addressing it through Figure 8a and Supplementary Video.

Fig 8a. Standing waves and frequency characteristics in VGLUT2 neurons during anesthesia. (a) Spatiotemporal patterns of waves Φ_G during anesthesia, at BSR level of 90%.

A complex picture of brain synchronization under anesthesia. Anesthesia can induce the brain to enter a more synchronized state, a slow-wave state similar to sleep. In this state, pancortical synchronized activity is dominant, as evidenced by our results as Φ_0 . However, we observed the presence of traveling waves other than these slow oscillations, suggesting a more complex synchronization state, which consist with earlier studies. Furthermore, although all slow oscillations (all Φ_0 under anesthesia) exhibit almost zero-lag synchronized activity, slow waves induced by different anesthetics display distinct characteristics in their spatial movement at small scale. As anesthetics typically affect the GABA system, this might be a comprehensive outcome, related to the mechanisms of synchronization including the interactions between GABAergic and glutamatergic neurons. This phenomenon warrants further investigation.

3. I could go on and on, there is a complete lack of understanding of the neurophysiological research on spontaneous task free activity, as e.g. evident in line 68 “..implying a close connection between DMN and burst suppression”. What does that mean?

Reply:

Thank you for pointing out. We removed expressions like this entirely.

4. Also on the developmental angle, certainly it is worthwhile studying developmental aspects in spontaneous activity, but the time points used are meaningless from a developmental viewpoint, as all time points are after eye opening (occurring around P 12), and after the GABA switch, so if they aim for studying these effects, they would have to study earlier time points.

Reply:

Thank you for your valuable advice. We agree that the effects of eye opening and GABA switch are valuable, and we are currently conducting these experiments. However, this experiment is very challenging. Young mice are difficult to implant with imaging windows, and they are very vulnerable to environmental and stimulation stress. The mortality rate of mice during imaging is rather high. We have conducted several pilot experiments to explore the experimental procedures and environmental settings. We have currently obtained experimental results from one mouse.

5. Lastly, also the cell-type specific analysis lack any understanding of the basic connectivity schemes of the cortex.

Reply:

We have revised the full Discussion to address the cell-type specific analysis and the connectivity schemes of the cortex, here's new text:

Cell-to-cell differences in movement impact spontaneous activity, although the main components are consistent with rest. We found traveling waves capture more details in the changes of neural activity compared to FC, and consist with subtype specificity found in studies before⁴⁶. All four types of neurons exhibit a decrease in wave quality, PV and VGLUT2 are the most affected, SOM is the least affected, and VIP is in the middle. Previous studies have shown that PV neurons split into distinct populations during movement: an excitatory group and a suppressed group⁴⁷. This functional dichotomy likely explains the dramatic decrease in wave quality we observed in PV neurons.

Different developmental trajectories in FC and traveling waves are observed for GABAergic interneurons and glutamatergic neurons.

FC of SOM and glutamatergic are established at an early stage, while FC of PV establishes connections throughout mouse development, and rapid expression of PV and establishment of synapses being among the most critical changes in the GABAergic system during this period⁴⁸. FC of VIP establishes after adolescence. Despite VIP expression being abundant during embryonic and early life stages⁴⁹, it remains unclear why its FC is mainly established between P28 and P56. VGLUT2 and SOM neurons establish relatively complete synchronizing structures at early stages. While VGLUT2 neurons exhibit nearly perfect time-lag synchronization even at early stages, GABAergic neurons except SOM lack this type of synchronization. Studies indicate that neural desynchronization may stem from shared input/output among excitatory and inhibitory neurons^{1,2}. The traveling wave robustness in VGLUT2 at early age presents a challenge in the mechanisms for time-lag synchronizations due to the immaturity of GABAergic neurons.

Interaction between different types of neurons. We observed opposite developmental trajectories between PV and VGLUT2 in traveling waves. It is known that PV expression and rapid synapse formation occur during this period⁴⁸ and PV inhibit glutamatergic neurons directly⁴⁶. The changes in VGLUT2 traveling waves from P14 to P56 may be associated with the establishment of PV synchronization. PV cells inhibit the cell bodies and proximal dendrites of pyramidal cells, while SOM cells target the distal dendrites⁵⁸⁻⁶⁰. Horizontal propagation in layer 5 pyramidal cells during the up state mainly occurs among their cell bodies, and thus, inhibition by PV cells at the cell bodies is likely the primary inhibitory system, with SOM cell inhibition of distal dendrites acting as a secondary system⁶¹. However, the impact of cell type on time-lag synchronization remains unclear. Notably, the timing of mature synchronization structures differs across cell types. After P28, VIP synchronization matures rapidly, while VGLUT2 synchronization exhibits a seemingly opposite pattern to its pre-P28 behavior. Before P28, traveling waves in VGLUT2 decrease coincides with the rise of that in PV. However, after P28, the traveling waves of VGLUT2 align with those of both PV and VIP, coinciding with modifications in the presynaptic terminal during this period⁶², resulting in a more complex picture. SOM neurons show early developed, stable synchronizing structures. In neural systems, excitatory activity typically triggers balanced inhibition for stability⁶³. VGLUT2 neurons rapidly mature during the early stages of eye opening, and SOM neurons form synaptic connections with excitatory neurons before P14, leading to early and relatively complete synchronizations similar to those seen in VGLUT2 neurons.

Reviewer #3 (Remarks to the Author):

In NCOMMS-23-37172, Shi et al investigate the neural underpinnings of so-called “resting-state waves” using calcium imaging with cell-type specificity (multiple cell types) in awake mice. The authors explored the spatiotemporal patterns emerging from such recordings under multiple conditions, including different anesthesia states, different developmental stages, nonrusting conditions, and also correlated extracted spatial patterns with gene expression. The authors verify that these waves underpin resting-state connectivity, and then also show that four different types of neurons (both excitatory and inhibitory) show similar global patterns. Interestingly, the developmental study showed varying timelines in the evolution of the waves for each cell type, which is a deep and interesting finding in my opinion. In non-resting conditions, waves were still decomposable and different to some extent compared with resting state waves.

Overall, this work represents a significant advance in our understanding of global brain activity patterns, which these days is becoming an increasingly “hot topic” (e.g. Pang et al, Nature 2023). In my view, the main advance here is the introduction of cell-type specificity in these measurements, which reveal the neural origins of these signals, as well as the developmental aspects of the study which I find both highly novel and intriguing in terms of the actual results. The study would certainly be of value to a broad audience and would appeal to Nat Comm readership.

I do have a few comments that I think would strengthen the work substantially:

Reply:

Thank you very much for the positive assessment of our work, as well as for your constructive and detailed feedback. We appreciate your highlighting of the remaining unclear points, which we address below:

Major Comments

1. Refinement of the novel aspects of the work

As mentioned above, there are many novel aspects to this work. However, I find some of the claims exaggerated. For instance, the subheading “Global structures of resting-state activity exist in mice” is a large overstatement and not per-se novel. These structures have been observed before, either indirectly by resting-state fMRI in mice (CAPs and QPPs for example, are “snapshots” of this kind of activity), and even in optical imaging experiments (e.g. Ma et al, Resting-state hemodynamics are spatiotemporally coupled to synchronized and symmetric neural activity in excitatory neurons, PNAS 2016 and papers thereafter). In addition, a recent study (Cabral J et al, Intrinsic macroscale oscillatory modes driving long range functional connectivity in female rat brains detected by ultrafast fMRI, Nature Communications 2023) revealed these waves in rats.

I think the authors could forego “forcing” novelty on the existence of the waves, and they probably should discuss (and cite) a few more papers that observed similar CAPs / QPPs and discuss their findings with relation to the prior art. This will not detract from the other impressive novel features presented in the work.

Reply:

We agree with the reviewer and have added more contextual information to the introduction. We have reviewed the entire manuscript and revised expressions that previously appeared exaggerated and which we had not fully considered. The subheading "Global structures of resting-state activity exist in mice" has now been changed to "Zero-lag and time-lag synchronization structures of spontaneous neural activity in VGLUT2 neurons."

2. Temporal aspects in the wave analyses

The authors did a great job in visualizing and comparing the spatial parts of the waves. However, it seems to me that the temporal aspects have been quite overlooked, which I believe is a large missed opportunity which would greatly enhance the study's relevance and impact.

The authors should deeply analyze the time series; they probably ought to observe damped oscillatory nature of these signals (c.f Cabral et al Nat Comm 2023), and so they could quantify frequency of activation along the different conditions (cell types, developmental stage), and further attempt to extract how frequently each of these patterns exist in each condition. This will reveal how often the brain "visits" these modes (so to speak).

I strongly recommend to look into the temporal aspects more deeply here.

Reply:

Thank you for your valuable advice. We employed damping analysis, as used in (Cabral et al., Nat Comm 2023), to analyze the temporal aspect of waves, as shown in Fig 1, 3f, 5f, and 8c-e. The method is sound, and we found results similar to those in (Cabral et al., Nat Comm 2023). Waves extracted by CPCA are decomposed based on their correlations, without considering frequency. However, the outcomes of this decomposition reveal frequency specificity, suggesting that the organization of long-range correlations between brain areas is frequency dependent. Below are the new figures and new texts in the Results section:

Fig. 1: Large-scale neural activity is organized by standing and traveling waves (VGLUT2, P56). (a) Fluorescent signals $S(t)$ of neural activity can be decomposed into a linear superposition of standing and travelling waves $\Phi_i(t)$ distributed throughout the cortex, which are extracted from spontaneous neural activity in VGLUT2 neurons at P56. (b) Spatiotemporal patterns (over one cycle) of waves $\Phi_0, \Phi_1, \Phi_4, \Phi_5$, showcasing only significant waves identified in the subsequent text. (c) Spatial distributions of waves $\Phi_0, \Phi_1, \Phi_4, \Phi_5$. ρ : Intensity distribution. θ : Phase distribution (time lag) in rad. (d) The power spectral density (PSD) of the fluorescence signal (pixel averaged, normalized to unit energy). (e) Spatial maps of spectral power in 6 non-overlapping frequency bands, with pixels normalized to unit energy. (f) Waveforms $\varphi_i(t)$ and autocorrelation functions of waves $\Phi_0, \Phi_1, \Phi_4, \Phi_5$. (g) Unit-energy PSDs of $\Phi_0, \Phi_1, \Phi_4, \Phi_5$, with inter-experiments variability (shaded). Nature frequencies (f) and damping ratios (ξ) of 10 strongest waves. (h) The proportion of the strongest 10 waves, measured by the variance ratio of each wave relative to the original signal. (i) The spatial distribution uniformity of the strongest 10 waves, measured by the circle variance of θ_i .

We found that the frequency distribution in space was uneven, with low frequencies dominating the entire cortex (Fig 1e). The frequency characteristics of the standing/traveling waves were different from those of the signals, with almost single-peaked distributions. This contrasts with the fluorescent signals, which have a broader frequency distribution. Natural frequency (f) reflects the frequency of energy concentration in the absence of damping. Damping ratio (ξ) measures the degree of energy concentration, with smaller values indicating higher concentration. It is also the reciprocal of the Q-factor ($\xi = 0.5Q^{-1}$). The natural frequencies for each standing/traveling wave, are similar in VGLUT2 neurons. The damping ratios ξ of these waves are low (Fig 1g), indicating a high degree of energy concentration in

a narrow band of frequencies. Waves extracted by CPCA are decomposed based on their correlations, without considering frequency. However, the results of the decomposition show frequency specificity, indicating that the organization of long-range correlations between brain areas is frequency-dependent, consistent with previous research results³³. However, Φ_0 , the “standing” wave, is special for a higher degree of inconsistency in frequency characteristics between different experiments and mice.

Fig. 3: Four types of neurons exhibit similar FC patterns at P56. (d) Variance explained by the top 10 waves in GABAergic and VGLUT2 neurons at P56. (e) Degree of similarity between waves from VGLUT2 and GABAergic neurons at P56, quantified by the minimum of mean squared errors (MSE) of correlation matrices from specific wave to all waves in VGLUT2. Lower MSE indicates greater similarity. (e) Nature frequencies (*f*) and damping ratios (ξ) of 10 strongest waves from four type of neurons. (f) The spatial distribution of similarity between the FC of different neuron types and that of VGLUT2 at P56, measured by Pearson coefficient of RSFC at each pixel.

Traveling waves reveal marked differences in cell-to-cell synchronization, and the extent of their similarity to VGLUT2 waves varies distinctly. The portions of waves in VGLUT2 are usually larger than those in GABAergic neurons (Fig 3d). Notably, waves in PV exhibits more similar to VGLUT2, while SOM shows less similar, with VIP falling between the two (Fig. 3e). Most waves in the four type of neurons occupy similar frequency bands and demonstrate comparable energy concentration within the frequency domain. However, some waves in GABAergic neurons, particularly those in SOM, exhibit slightly higher frequencies compared to those in VGLUT2 neurons and the frequency deviation of waves in GABAergic neurons often surpasses that of VGLUT2 (Fig. 3f).

Fig. 5: Spatial waves vary during development. (f-g) Nature frequencies (f) and damping ratios (ξ) of 10 strongest waves from four type of neurons at P14, P28 respectively.

The frequency characteristics of waves in PV, SOM and VIP neurons in young mice are at a higher frequency than VGLUT2 and adults, and are relatively chaotic, varying between different experiments (Fig 3f, Fig 5f,g). VGLUT2 neurons are relatively more consistent with adults. This suggests that frequency response characteristics of GABAergic neurons are adjusted during development to match those of excitatory neurons.

Fig. 8: Standing waves and frequency characteristics in VGLUT2 neurons during anesthesia. (b) Prominent co-activation patterns in anesthetized state (90% BSR). (c-e) Nature frequencies (f) and damping ratios (ξ) of 10 strongest waves from VGLUT2 neurons at BSR level 50%, 70% and 90%.

During anesthesia, the frequency characteristics of the waves are higher than in the awake state and exhibit a relatively chaotic nature, varying between different experiments, as shown in Fig. 8c-e.

3. Comparison of not-necessarily overlapping wave patterns

The authors compare waves across different conditions. But it seems like waves have varying spatiotemporal characteristics (e.g, phi1 in condition A could be different from phi1 in condition B; or, there could be a different number of underlying waves contributing to the total signal between conditions). How was this taken into account?

Reply:

We compared the correlation matrices of all waves from various cell types in different states with Φ_0 , Φ_1 , Φ_2 , Φ_4 and Φ_5 from VGLUT2 at P56. We identified those with the highest similarity as belonging to the same category. New text in Method, and Supplementary Fig 3:

There is no direct method to compare different waves from CPCA results. To identify corresponding waves, we examine the connectivity matrices of the waves (C_{Φ_i}). We start with the spatiotemporal pattern of each wave and compute its connectivity matrix, the correlation coefficients between pixels ($C_{\Phi_i}(j, k) = Pearson([S_R^j(t), t = 1, 2, \dots], [S_R^k(t), t = 1, 2, \dots])$), $S_R^j(t)$ is the signal of pixel j reconstructed from wave Φ_i . We then compare these connectivity matrices based on mean square error (MSE), considering the waves with the lowest MSE as the same type. This mean square error is shown in Supplementary Fig 3.

Supplementary Fig 3. **Correlation matrices of reconstructed time courses from standing/traveling waves.** (a-e) Correlation matrices of reconstructed time courses from Φ_G , Φ_{SM1} , Φ_{SM2} , Φ_T and $\Phi_{DMN-like}$. We compared the correlation matrices of all waves from various cell types in different states with Φ_0 , Φ_1 , Φ_2 , Φ_4 and Φ_5 from VGLUT2 at P56. We identified those with the highest similarity as belonging to the same category. The figure displays the correlation matrices of the waves with the highest similarities. The text below each matrix indicates the wave and its similarity to the corresponding wave in VGLUT2, measured by mean square error (MSE), with lower values indicating better similarity.

4. Unclear figures and missing statistical comparisons

4a. There are several figures that are difficult to understand. Most prominently, perhaps, Fig 7c: what are conditions 1-12? 12 waves? This should really be spelled out.

4b. In addition, Fig 7a, the dashed lines are confusing – what do they represent?

4c. Fig 2c should be explained in the text.

4d. There are no error bars on Fig 2b, 2d and 2f, and statistical significance is not indicated. Please quantify.

4e. How were multiple comparisons accounted for in this work?

4f. Add heatmap to Fig 4b

4g. In Fig 1, is phi3 deliberately omitted?

4h. Figure 4c and 4d: why are there two of every line color? This figure needs to be explained much better in the text and in the caption.

Reply:

A, F: In the revised version, we have substantially improved the presentation of our results and thoroughly reviewed the entire article to ensure that each figure, is explicitly explained within the text.

B: Changes in namely Φ_{SM2} , Φ_T and $\Phi_{DMN-like}$ of four neuron types at three developmental stages (post GSR). Dashed lines show the changes in the corresponding wave proportions during development, and images show the changes in the corresponding wave spatial distributions. (Add new legend)

C: We removed Fig 2c to make the manuscript more focus.

D, E: Regarding statistics and multiple testing corrections, we have employed the False Discovery Rate (FDR) method (Benjamini-Hochberg algorithm). The details of this approach are now included both in the methods section and in the legends of each figure.

To account for multiple testing and control the False Discovery Rate (FDR), we corrected the p-values using the Benjamini-Hochberg algorithm⁶¹. We considered p-values less than 0.05 as statistically significant.

G: We calculated 10 waves for each state, all of which are presented in the supplementary figures. We chose to display a few particularly significant waves in Fig 1, which are referenced later in the text. (Add new legend)

H: Trends of the average FC at 5 seed points during development. The positive connections and negative connections are averaged separately, shown in top and bottom panel respectively. The error bars show the standard deviation of FC between each mouse. (d) Area size of regions with significant connections at 5 seed points.

5. Sensitivity of different cell-type markers

Do the authors expect any differences in sensitivity with each cell-type calcium reporter? Could differing expression patterns/efficiency of each marker have played a role in some of the findings in the work?

Reply:

In the revised version, we have included a new Supplementary Fig 4g, which illustrates the differences in fluorescence intensity across various cell types and states. Additionally, we conducted simulation experiments to investigate the impact of differing expression patterns and efficiencies, as detailed in Supplementary Fi 5. Our findings demonstrate that the efficiency of expression has minimal effect on our methodology, affirming the robustness of our approach.

Supplementary Fig. 4: Fluorescence microscopy imaging in vivo and data processing. (g) Fluorescence intensity in different types of mice and conditions. The differences are come from sensitivity with each cell-type calcium reporters.

Supplementary Fig. 5: CPCA analysis using simulated data. (a-c) The results of three CPCA analyses using simulated data. In these three simulations, we simulated fluorescent proteins with different efficiencies and of different spatial distributions, as shown in the figure of simulated efficiency.

We normalized each pixel prior to conducting CPCA. This normalization allowed for comparability of signal magnitudes across different FC experiments and locations. However, the effectiveness of this method needed to be demonstrated. To investigate the differences in the spatial distribution and efficiency of fluorescent protein expression, we conducted simulation experiments (Supplementary Fig 5). We generated three sets of unrelated signals and created three groups of spatial amplitude and time delay distributions, also adding zero-mean Gaussian noise. To mimic variations in fluorescent protein expression and spatial distribution, we produced several spatial distribution patterns (simulated efficiency in Supplementary Fig 5). The generated signals were linearly superimposed and coupled with zero-mean

Gaussian noise and simulated fluorescent protein expression spatial distributions to create simulated neural activity signals, which were then subjected to CPCA. We conducted three sets of simulations with different sizes and spatial distributions of simulated efficiency. The results indicated that differing expression patterns/efficiency of each marker did not significantly affect the outcomes of CPCA.

6. Connection with fMRI

The impact of the work here could be dramatically enhanced if the authors analyzed their data convolved with a hemodynamic response function, and provided insight into rs-fMRI studies from the four cell types they recorded. Something akin to what Ma et al PNAS 2016 did would really boost the impact of this work, especially if discussed in connection with CAPs, QPPs and Intrinsic Oscillatory Modes.

Reply:

Thank you for your suggestions. We agree that the HRF of different cell types are valuable for the field, and we are currently conducting these experiments. We are mainly focusing on supplementing the experiments of the developmental process at a much early age. For the HRF experiment, we have built an intrinsic optical imaging system and conducted preliminary experiments. The preliminary results are as follows:

Minor Comments

6. The paper could benefit from heavy editing. The Introduction section is (paradoxically) both not concise and incomplete.

7. In addition, the text from line 61 to the end of the introduction, is actually a mini results section. It should be heavily edited and perhaps even completely removed.

Reply to 6, 7:

Thank you for your advice. We have significantly improved the expression of the entire article, including the Introduction, and removed the text from line 61 to the end of the introduction. As the reviewers can find, the Introduction section has almost been rewritten to make it concise and complete.

8. Line 107: it would be probably worthwhile to adhere to already accepted nomenclature such as “QPP” or “intrinsic oscillatory mode”.

Reply:

Thanks for the advice. We have connected our work with other methods, including QPP and the work of (Cabral et al. Nat Comm 2023).

These waves are widely distributed across the cortex and recur in a quasi-periodic manner. This characteristic of quasi-periodic recurrence is similar to the Quasi-periodic patterns (QPPs)³² or intrinsic oscillatory mode³³ found in fMRI studies.

9. Figure 2f, the yellow legend for VGLUT2 slipped downward and is overlapping with the green SOM legend. Please fix.

Reply:

We have redrawn Fig 2, and fixed typesetting problems of this type.

10. Line 148: "... having exhibiting average" – unfinished sentence.

Reply:

Thanks for pointing out, fixed.

11. Fig.4 – why not add P14->56?

Reply:

P14->56 is now in Supplementary Figure 2, because the image is too large to fit on one page, so it is placed in the supplementary figure.

Supplementary Fig. 2: Similarity between FC and standing/traveling waves. (a) The spatial distribution of similarity between the RSFC of different neuron types and that of VGLUT2 at P56 (RSFC similarity), measured by Pearson coefficient of RSFC at each pixel. The contour illustrates the spatial distribution of Φ_1 . (b) The violin plot of RSFC similarity. (c) Changes of seed-seed FC post GSR in four types of neurons between P14 and P56, with diagonal entries representing the short-range FC changes ($p < 0.05$, Fisher's z-transformed t-tests, after FDR correction with threshold of 0.05).

12. RSW is a somewhat paradoxical name as you find them under activity too. Perhaps go with Intrinsic Macroscale Oscillatory Modes like Cabral et al 2023?

Reply:

Thank you a lot for your suggestion. We agree with the reviewer's point of view. For patterns that move in space, such as the results of CPCA decomposition, most studies use names such as propagating waves, standing and traveling waves, and time-lag synchronize, as shown in the introduction. Therefore, following these works, we have changed the name to standing and traveling waves.

13. Lines 319-322: this is arguably one of the most important findings of this work. Suggest expanding on this.

Reply:

We have extended the Discussion section for this:

Different developmental trajectories in FC and traveling waves are observed for GABAergic interneurons and glutamatergic neurons. FC of SOM and glutamatergic are established at an early stage, while FC of PV establishes connections throughout mouse development, and rapid expression of PV and establishment of synapses being among the most critical changes in the GABAergic system during this period⁴⁸. FC of VIP establishes after adolescence. Despite VIP expression being abundant during embryonic and early life stages⁴⁹, it remains unclear why its FC is mainly established between P28 and P56. VGLUT2 and SOM neurons establish relatively complete synchronizing structures at early stages. While VGLUT2 neurons exhibit nearly perfect time-lag synchronization even at early stages, GABAergic neurons except SOM lack this type of synchronization. Studies indicate that neural desynchronization may stem from shared input/output among excitatory and inhibitory neurons^{1,2}. The traveling wave robustness in VGLUT2 at early age presents a challenge in the mechanisms for time-lag synchronizations due to the immaturity of GABAergic neurons.

Interaction between different types of neurons. We observed opposite developmental trajectories between PV and VGLUT2 in traveling waves. It is known that PV expression and rapid synapse formation occur during this period⁴⁸ and PV inhibit glutamatergic neurons directly⁴⁶. The changes in VGLUT2 traveling waves from P14 to P56 may be associated with the establishment of PV synchronization. PV cells inhibit the cell bodies and proximal dendrites of pyramidal cells, while SOM cells target the distal dendrites⁵⁸⁻⁶⁰. Horizontal propagation in layer 5 pyramidal cells during the up state mainly occurs among their cell bodies, and thus, inhibition by PV cells at the cell bodies is likely the primary inhibitory system, with SOM cell inhibition of distal dendrites acting as a secondary system⁶¹. However, the impact of cell type on time-lag synchronization remains unclear. Notably, the timing of mature synchronization structures differs across cell types. After P28, VIP synchronization matures rapidly, while VGLUT2 synchronization exhibits a seemingly opposite pattern to its pre-P28 behavior. Before P28, traveling waves in VGLUT2 decrease coincides with the rise of that in PV. However, after P28, the traveling waves of VGLUT2 align with those of both PV and VIP, coinciding with modifications in the presynaptic terminal during this period⁶², resulting in a more complex picture. SOM neurons show early developed, stable synchronizing structures. In neural systems, excitatory activity typically triggers balanced inhibition for stability⁶³. VGLUT2 neurons rapidly mature during the early stages of eye opening, and SOM neurons form synaptic connections with excitatory neurons before P14, leading to early and relatively complete synchronizations similar to those seen in VGLUT2 neurons.

14. line 335: “tempo spatial” is a misnomer. Either temporospatial or spatiotemporal. Also, at this place, it would be good to connect to Cabral et al Nat Comm 2023, with their ultrafast acquisitions.

Reply:

Thanks for the advice. We have fixed the typo and connected our work with the work of (Cabral et al. Nat Comm 2023).

15. Lines 342-346: too speculative. Suggest removing altogether

Reply:

Removed as required.

REVIEWERS' COMMENTS

Reviewer #1 (Remarks to the Author):

The manuscript has been revised to address some of the concerns raised in my previous review. However, there remain multiple errors (for example, at lines 80-81, 182-183, 303-304, 384 and within Figure 7a) that should be addressed to facilitate a comprehensive evaluation. Moreover, the order of figure panel citations in the text could be better sequenced to enhance readability. I recommend a detailed proofreading of the manuscript to resolve these issues.

The manuscript provides a detailed exploration of developmental changes in cortex-wide activity across different inhibitory neurons, which is noteworthy. However, considering the existing body of literature, the study's contributions to the field could be viewed as marginal. Although the experiments and analyses were performed well, I maintain some reservations regarding the study's provision of novel mechanistic insights into the interplay of various cell types in shaping cortex-wide neural activity.

Reviewer #1 (Remarks on code availability):

I have not reviewed the code.

Reviewer #2 (Remarks to the Author):

The authors have addressed my main concerns and they have drastically improved the manuscript. They put in considerable efforts, and they do clearly state the limitations of their study, particularly in terms of the developmental axis. I am still not convinced that the terminology of "burst suppression" is fitting best, but I do follow their argument, that indeed in the clinical field, this is a still widely used terminus. And as the authors do acknowledge and describe the relation of burst suppression with slow oscillations, I do support publication of the manuscript.

Reviewer #3 (Remarks to the Author):

The authors have addressed all the scientific concerns I have raised in my previous review. The scientific content has improved further (and it was already excellent before), and I believe that this work will be of

high interest and significance given its depth and insights on the synchronizing structures in different cell types. Just a few minor comments remain:

1. The English level throughout the manuscript is still quite poor, and there are many linguistic oddities and typos to correct. They are too numerous to list here, and the authors would do well to improve dramatically on this aspect.

2. In Fig 1f, please add a timescale so that the reader can appreciate the duration of the window of acquisition. I also suggest in all relevant figures to replace the time domain unit from [frame] to seconds as this is clearer.

3. Is “Nature Frequency” a common/accepted term? (not to my knowledge). Do you mean the natural frequency or rather the resonance frequency?

Reviewer #3 (Remarks on code availability):

I am not a python user so I did not test this code.

We thank the reviewers for their thoughtful, constructive and detailed feedback on our manuscript. We have edited the main text in response to reviewer comments (as detailed below); these changes are **highlighted in red** in the revised manuscript and the response. In our point-by-point response to reviewers below, the reviewers' remarks are presented in **blue** and our responses are shown in black text.

Reviewer #1 (Remarks to the Author):

The manuscript has been revised to address some of the concerns raised in my previous review. However, there remain multiple errors (for example, at lines 80-81, 182-183, 303-304, 384 and within Figure 7a) that should be addressed to facilitate a comprehensive evaluation. Moreover, the order of figure panel citations in the text could be better sequenced to enhance readability. I recommend a detailed proofreading of the manuscript to resolve these issues.

The manuscript provides a detailed exploration of developmental changes in cortex-wide activity across different inhibitory neurons, which is noteworthy. However, considering the existing body of literature, the study's contributions to the field could be viewed as marginal. Although the experiments and analyses were performed well, I maintain some reservations regarding the study's provision of novel mechanistic insights into the interplay of various cell types in shaping cortex-wide neural activity.

Reviewer #1 (Remarks on code availability):

I have not reviewed the code.

Reply:

Thank you very much for the constructive and detailed comments along the way – they really improved the work. We reviewed the text and fixed typos like lines 80-81. The line breaks within Fig 7a indicate a change from/to 0, for which we have revised the figure legend to explain this. Considering the existing literature, our findings indicate a link between traveling waves and long-range functional connectivity/resting-state networks (RSNs) from cellular and developmental perspectives, which is meaningful for interpretations of FC and RSNs.

Reviewer #2 (Remarks to the Author):

The authors have addressed my main concerns and they have drastically improved the manuscript. They put in considerable efforts, and they do clearly state the limitations of their study, particularly in terms of the developmental axis. I am still not convinced that the terminology of "burst suppression" is fitting best, but I do follow their argument, that indeed in the clinical field, this is a still widely used term. And as the authors do acknowledge and describe the relation of burst suppression with slow oscillations, I do support publication of the manuscript.

Reply:

Thank you very much for the time you spent on reading and for all the insightful comments along the way – they really improved the work. Thank you.

Reviewer #3 (Remarks to the Author):

The authors have addressed all the scientific concerns I have raised in my previous review. The scientific content has improved further (and it was already excellent before), and I believe that this work will be of high interest and significance given its depth and insights on the synchronizing structures in different cell types. Just a few minor comments remain:

Reply:

Thank you very much for the time you spent on reading and for all the insightful comments along the way – they really improved the work. Thank you.

1. The English level throughout the manuscript is still quite poor, and there are many linguistic oddities and typos to correct. They are too numerous to list here, and the authors would do well to improve dramatically on this aspect.

Reply:

Thank you for the advice. We've consulted an English language editing service to improve the clarity of the manuscript, and we believe it's much better now.

2. In Fig 1f, please add a timescale so that the reader can appreciate the duration of the window of acquisition. I also suggest in all relevant figures to replace the time domain unit from [frame] to seconds as this is clearer.

Reply:

Thank you for the advice. We've added a timescale in Fig 1f and replaced all time domain unit from frame to seconds.

3. Is "Nature Frequency" a common/accepted term? (not to my knowledge). Do you mean the natural frequency or rather the resonance frequency?

Reply:

Thank you for pointing out. It's a typo for natural frequency. We've fixed these typos.

Reviewer #3 (Remarks on code availability):

I am not a python user so I did not test this code.